# Interpretable protein-DNA interactions captured by structure-sequence optimization

Yafan Zhang[1], Irene Silvernail[2], Zhuyang Lin[1], Xingcheng Lin[1,2]*

[1]Bioinformatics Research Center, North Carolina State University, Raleigh, United States; [2]Department of Physics, North Carolina State University, Raleigh, United States

## eLife Assessment

This **valuable** work presents an interpretable protein-DNA Energy Associative (IDEA) model for predicting binding sites and affinities of DNA-binding proteins. While the method is **convincing**, it requires some adaptation for application to different proteins. The IDEA method is available and can be potentially used for predicting genome-wide protein-DNA binding sites.

**Abstract** Sequence-specific DNA recognition underlies essential processes in gene regulation, yet methods for simultaneous predictions of genomic DNA recognition sites and their binding affinity remain lacking. Here, we present the Interpretable protein-DNA Energy Associative (IDEA) model, a residue-level, interpretable biophysical model capable of predicting binding sites and affinities of DNA-binding proteins. By fusing structures and sequences of known protein-DNA complexes into an optimized energy model, IDEA enables direct interpretation of physicochemical interactions among individual amino acids and nucleotides. We demonstrate that this energy model can accurately predict DNA recognition sites and their binding strengths across various protein families. Additionally, the IDEA model is integrated into a coarse-grained simulation framework that quantitatively captures the absolute protein-DNA binding free energies. Overall, IDEA provides an integrated computational platform that alleviates experimental costs and biases in assessing DNA recognition and can be utilized for mechanistic studies of various DNA-recognition processes.

*For correspondence:
Xingcheng_Lin@ncsu.edu

## Introduction

Gene regulation is essential for controlling the timing and extent of gene expression in cells. It guides important processes from development to disease response (*Lee and Young, 2013*). Gene expression is tightly controlled by various DNA-binding proteins, including transcription factors (TFs) and epigenetic regulators (*Orphanides and Reinberg, 2002*; *Ren et al., 2000*; *Jaenisch and Bird, 2003*). TFs initiate gene expression by binding to specific DNA sequences, during which time a primary RNA transcript is synthesized from a gene's DNA (*Latchman, 1997*). On the other hand, epigenetic regulators control gene expression by binding to specific chromatin regions, which spread post-translational modifications that modulate 3D genome organization (*Owen et al., 2023*). Therefore, characterizing the DNA-interaction specificities of DNA binding proteins is critical for understanding the molecular mechanisms underlying many DNA-templated processes (*Stormo and Zhao, 2010*).

To establish a comprehensive understanding of DNA binding processes, various experimental technologies have been developed and utilized (*Solomon et al., 1988*; *Park, 2009*; *Bartlett et al., 2017*; *Berger and Bulyk, 2009*; *Meng et al., 2005*; *Maerkl and Quake, 2007*; *Fordyce et al., 2010*;

*Ogawa and Biggin, 2012*; *Isakova et al., 2017*). These methods, such as Chromatin Immunoprecipitation followed by sequencing (ChIP-Seq; *Solomon et al., 1988*; *Park, 2009*; *Furey, 2012*), protein-binding microarray (PBM; *Bulyk et al., 2001*; *Gordân et al., 2013*; *Afek et al., 2020*), and systematic evolution of ligands by exponential enrichment (SELEX; *Jolma et al., 2010*; *Ogawa and Biggin, 2012*; *Isakova et al., 2017*), have proven invaluable for measuring DNA-specific protein recognition. Nonetheless, due to the need for protein-specific antibodies (*Furey, 2012*), as well as the cost and intrinsic bias associated with these experiments (*Kohlberger and Gadermaier, 2022*), high-throughput (HT) measurement of large numbers of protein-DNA variants remains challenging.

Computational methods complement experimental efforts by providing the initial filter for assessing sequence-specific protein-DNA binding affinity. Numerous methods have emerged to enable predictions of binding sites and affinities of DNA-binding proteins (*Weirauch et al., 2013*; *Alipanahi et al., 2015*; *Zhou et al., 2015*; *Yang et al., 2017*; *Rastogi et al., 2018*; *Roche et al., 2024*; *Yang et al., 2023*; *Nguyen et al., 2019*; *Rube et al., 2022*). These methods often utilized machine-learning-based training to extract sequence preference information from DNA or protein by utilizing experimental HT assays (*Weirauch et al., 2013*; *Alipanahi et al., 2015*; *Zhou et al., 2015*; *Yang et al., 2017*; *Rastogi et al., 2018*; *Rube et al., 2022*), which rely on the availability and quality of experimental binding assays. Additionally, many approaches employ deep neural networks (*Roche et al., 2024*; *Liu and Tian, 2023*; *Nguyen et al., 2019*), which could obscure the interpretation of interaction patterns governing protein-DNA binding specificities. Understanding these patterns, however, is crucial for elucidating the molecular mechanisms underlying various DNA-recognition processes, such as those seen in TFs (*Siggers and Gordân, 2014*).

Nowadays, over 5000 protein-DNA 3D structures, including TF-DNA complexes, have been published (*Mitra et al., 2025*; *Harini et al., 2022*). These data provide invaluable resources for understanding the physicochemical properties of protein-DNA binding patterns, extending beyond mere sequence information. Utilization of these data enables the training of a model for predicting the binding affinities or specificities of protein-DNA interactions. Emerging deep learning (*Mitra et al., 2024*), probabilistic (*Wetzel et al., 2022*), and chemistry-based models (*Chiu et al., 2023*) are beginning to learn from protein-DNA co-crystal structures. Their predictions are biophysically meaningful and can facilitate mechanistic understanding of molecular interactions, thus holding great potential to guide experimental design and synthetic biology applications. The very robustness of evolution (*Bryngelson and Wolynes, 1987*; *Onuchic et al., 1997*; *Schafer et al., 2014*; *Chu et al., 2021*) provides an opportunity to extract the sequence-structure relationships embedded in existing complexes. Guided by this principle, we can learn an interpretable binding energy landscape that governs the recognition processes of DNA-binding proteins.

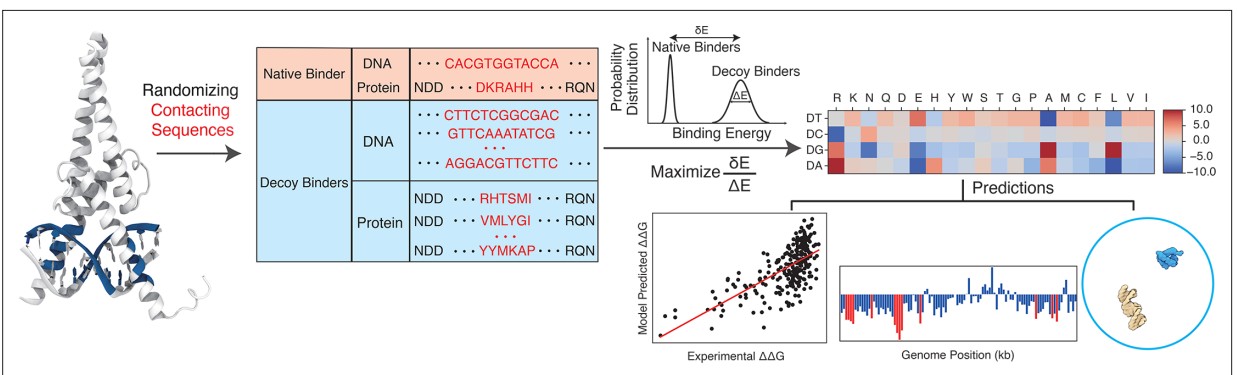

**Figure 1.** Overview of the IDEA protocol. The protein-DNA complex, represented by the human MAX-DNA complex structure (PDB ID: 1HLO), was used for training the IDEA model. The sequences of the protein and DNA residues that form close contacts (highlighted in blue on the structure) in the structure were included in the training dataset. In addition, a series of synthetic decoy sequences was generated by randomizing the contacting residues in both the protein and DNA sequences. The amino acid-nucleotide energy model was then optimized by maximizing the ratio of the binding energy gap ($\delta E$) between protein and DNA in the native complex and the decoy complexes relative to the energy variance ($\Delta E$). The optimized energy model can be used for multiple predictive applications, including the evaluation of binding free energies for various protein-DNA sequence pairs, prediction of genomic DNA binding sites by transcription factors or other DNA-binding proteins, and integration into a sequence-specific, residue-resolution simulation framework for dynamic simulations.

Here, we introduce the Interpretable protein-DNA Energy Associative (IDEA) model, a predictive model that learns protein-DNA physicochemical interactions by fusing available biophysical structures and their associated sequences into an optimized energy model (*Figure 1*). We show that the model can be used to accurately predict the sequence-specific DNA binding affinities of DNA-binding proteins and is transferrable across the same protein superfamily. Moreover, the model can be enhanced by incorporating experimental binding data and can be generalized to enable base-pair resolution predictions of genomic DNA-binding sites. Notably, IDEA learns a family-specific interaction matrix that quantifies energetic interactions between each amino acid and nucleotide, allowing for a direct interpretation of the 'molecular grammar' governing sequence-specific protein-DNA binding affinities. This interpretable energy model is further integrated into a simulation framework, enabling mechanistic studies of various biomolecular functions involving protein-DNA dynamics.

## Results

### Predictive protein-DNA energy model at residue resolution

IDEA is a coarse-grained biophysical model at the residue resolution for investigating protein-DNA binding interactions (*Figure 1*). It integrates both structures and corresponding sequences of known protein-DNA complexes to learn an interpretable energy model based on the interacting amino acids and nucleotides at the protein-DNA binding interface. The model is trained using available protein-DNA complexes curated from existing databases (*Burley et al., 2023*; *Mitra et al., 2025*). Unlike existing deep-learning-based protein-DNA binding prediction models, IDEA aims to learn a physicochemical-based energy model that quantitatively characterizes sequence-specific interactions between amino acids and nucleotides, thereby interpreting the 'molecular grammar' driving the binding energetics of protein-DNA interactions. The optimized energy model can be used to predict the binding affinity of any given protein-DNA pair based on its structures and sequences. Additionally, it enables the prediction of genomic DNA binding sites by a given protein, such as a TF. Finally, the learned energy model can be incorporated into a simulation framework to study the dynamics of DNA-binding processes, revealing mechanistic insights into various DNA-templated processes. Further details of the optimization protocol are provided in the Materials and methods section *Energy model optimization*.

### IDEA accurately predicts sequence-specific protein-DNA binding affinity

We first examine the predictive accuracy of IDEA by comparing its predicted TF-DNA binding affinities with experimental measurements (*Maerkl and Quake, 2007*; *Fordyce et al., 2010*). We focused on the MAX TF, a basic Helix-loop-helix (bHLH) TF with the most comprehensive available experimental binding data. The binding affinity of MAX to various DNA sequences has been quantified by multiple experimental platforms, including different SELEX variants (*Jolma et al., 2010*; *Ogawa and Biggin, 2012*; *Isakova et al., 2017*) and microfluidic-based MITOMI assay (*Maerkl and Quake, 2007*; *Geertz et al., 2012*). Among them, MITOMI quantitatively measured the binding affinities of MAX TF to a comprehensive set of 255 DNA sequences with mutations in the enhancer box (E-box) motif, the consensus MAX DNA binding sequence. This dataset serves as a reference for us to benchmark our model predictions. Our de novo prediction, based on one MAX crystal structure (PDB ID: 1HLO), correlates well with the experimental values (Pearson correlation coefficient 0.67) and correctly ranks the binding affinities of those DNA sequences (Spearman's rank correlation coefficient 0.65, *Figure 2A*). Including additional human-associated MAX-DNA complex structures and their associated sequences in the model training slightly improves the predictions (Pearson correlation coefficient 0.68, Spearman's rank correlation coefficient 0.65, *Figure 2—figure supplement 1*), albeit not significantly, likely due to the high structural similarity of the protein-DNA interfaces across all MAX proteins. Prior works have proven it instrumental in incorporating experimental binding data to improve predicted protein-nucleic acid binding affinities (*Alipanahi et al., 2015*; *Rastogi et al., 2018*; *Zhou et al., 2018*). Inspired by these approaches, we developed an enhanced predictive protocol that integrates additional experimental binding data if available (see the Materials and methods section *Enhanced modeling prediction with SELEX data*). Encouragingly, when training the model with the SELEX-seq data (*Rastogi et al., 2018*) of MAX TF, IDEA showed a significant improvement in

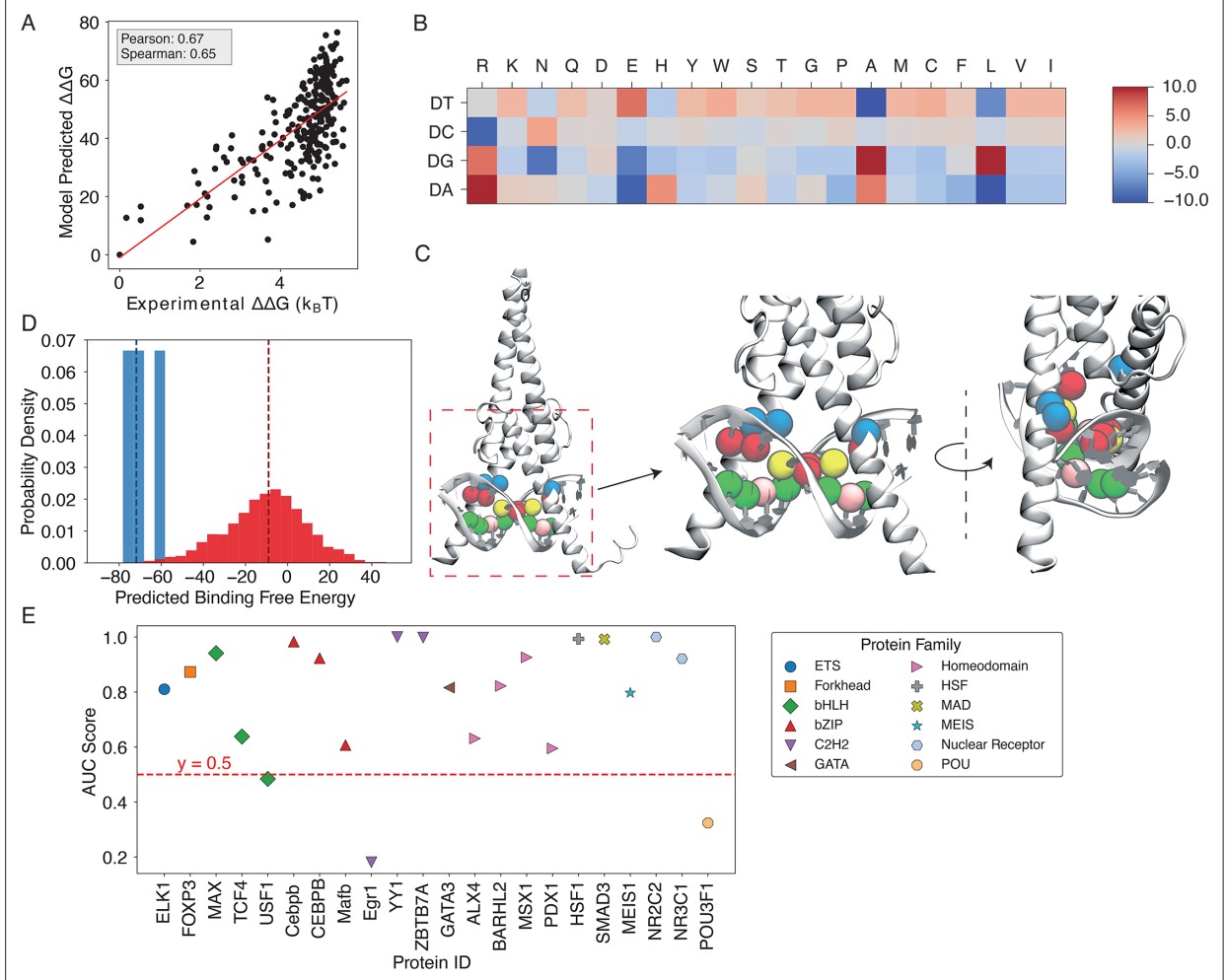

**Figure 2.** Results for MAX-based predictions. (**A**) The binding free energies calculated by IDEA, trained using a single MAX-DNA complex (PDB ID: 1HLO), correlate well with experimentally measured MAX-DNA binding free energies (*Maerkl and Quake, 2007*). $\Delta\Delta G$ represents the changes in binding free energy relative to that of the wild-type protein-DNA complex. (**B**) The heatmap, derived from the optimized energy model, illustrates key amino acid-nucleotide interactions governing MAX-DNA recognition, showing pairwise interaction energies between 20 amino acids and the four DNA nucleotide—DA (deoxyadenosine), DT (deoxythymidine), DC (deoxycytidine), and DG (deoxyguanosine). Both the predicted binding free energies and the optimized energy model are expressed in reduced units, as explained in the Materials and methods section *Training protocol*. Each cell represents the optimized energy contribution, where blue indicates more favorable (lower) energy values, and red indicates less favorable (higher) values. (**C**) The 3D structure of the MAX-DNA complex (zoomed in with different views) highlights key amino acid-nucleotide contacts at the protein-DNA interface. Notably, several DNA deoxycytidines (red spheres) form close contacts with arginines (blue spheres). Additional nucleotide color coding: adenine (yellow spheres), guanine (green spheres), thymine (pink spheres). (**D**) Probability density distributions of predicted binding free energies for strong (blue) and weak (red) binders of the protein ZBTB7A. The median of each distribution is marked with a dashed line. (**E**) Summary of AUC scores for protein-DNA pairs across 12 protein families, calculated based on the predicted probability distributions of binding free energies.

The online version of this article includes the following source data and figure supplement(s) for figure 2:

**Figure supplement 1.** Including additional human Max-DNA complex structures in IDEA training improves prediction.

**Figure supplement 2.** Enhanced IDEA prediction by integrating SELEX-seq data for MAX transcription factor.

**Figure supplement 3.** Refined energy model with SELEX data integration reveals additional physicochemical insights into protein-DNA interactions.

**Figure supplement 4.** Evaluation of IDEA's predictive accuracy for distinguishing strong from weak protein-DNA binding interactions.

**Figure supplement 5.** Summary of balanced PRAUC scores for protein-DNA pairs across 12 protein families.

**Figure supplement 6.** Performance comparison of the IDEA model with other prediction methods.

**Figure supplement 6—source data 1.** Raw AUC scores for distinguishing strong and weak binders across 22 proteins from 12 protein families using six different prediction methods: IDEA, IDEA augmented with binding sequences (IDEA-seq), ProBound, DeepBind, DBD-Hunter, and rCLAMPS.

**Figure supplement 6—source data 2.** Raw PRAUC scores for distinguishing strong and weak binders across 22 proteins from 12 protein families using

*Figure 2 continued on next page*

*Figure 2 continued*

six different prediction methods: IDEA, IDEA augmented with binding sequences (IDEA-seq), ProBound, DeepBind, DBD-Hunter, and rCLAMPS.

**Figure supplement 7.** IDEA outperforms other predictors in cross-validation analysis of protein-DNA binding affinity.

**Figure supplement 8.** IDEA correctly predicts the protein-DNA recognition by additional transcription factors.

**Figure supplement 9.** Enhanced predictive accuracy with the inclusion of Zif268 and related CATH protein structures.

reproducing the MITOMI measurement (Pearson correlation coefficient 0.79, Spearman's rank correlation coefficient 0.76, *Figure 2—figure supplement 2*).

## IDEA decodes the molecular grammar governing protein-DNA interactions

IDEA protocol learns the physicochemical interactions that determine protein-DNA interactions by utilizing the sequence-structure relationship embedded in the protein-DNA experimental structures. Such a physicochemical interaction pattern can be interpreted directly from the learned energy model. To illustrate this, we focused on the energy model learned from the MAX-DNA complexes (*Figure 2B*). Notably, DNA deoxycytidine (DC) exhibited strong interactions with protein arginine (R), consistent with the fact that the E-box region (CACGTG) frequently attracts the positively charged residues of bHLH TFs (*Blackwell et al., 1990*). Importantly, arginine was in close contact with deoxycytidine in the crystal structure (*Figure 2C*), thus consistent with the strong DC-R interactions shown in the learned energy model. Upon integrating the SELEX data into our model training, we found that the improved energy model shows additional unfavorable interactions between protein glutamic acid (E) and deoxycytidine, consistent with their negative charges (*Figure 2—figure supplement 3*). Thus, including more experimental data can boost IDEA's predictive accuracy by refining the amino-acid-nucleotide interacting energy model to better align with physical principles.

## IDEA generalizes across various protein families

To examine IDEA's predictive accuracy across different DNA-binding protein families, we applied it to calculate protein-DNA binding affinities using a comprehensive HT-SELEX dataset (*Yang et al., 2017*). We focused on evaluating the capability of IDEA to distinguish strong binders from weak binders for each protein with an experimentally determined structure. We calculated the probability density distribution of the top and bottom binders identified in the SELEX experiment. A well-separated distribution indicates the successful identification of strong binders by IDEA (*Figure 2*; *Figure 2—figure supplement 4*). Receiver operating characteristic (ROC) and precision-recall (PR) analyses were performed to calculate the AUC and the precision-recall AUC (PRAUC) scores for these predictions. Further details are provided in the Materials and methods section *Evaluation of IDEA prediction using HT-SELEX data*. Our analysis shows that IDEA successfully differentiates strong from weak binders for 80% of the 22 proteins across 12 protein families, achieving AUC and balanced PRAUC scores greater than 0.5 (*Figure 2*, *Figure 2—figure supplement 5*). To benchmark IDEA's performance against other leading methods, we compared its predictions with several popular models, including the sequence-based predictive models ProBound (*Rube et al., 2022*) and DeepBind (*Alipanahi et al., 2015*), the family-based energy model rCLAMPS (*Wetzel et al., 2022*), and the knowledge-based energy model DBD-Hunter (*Gao and Skolnick, 2008*). IDEA demonstrates performance comparable to these state-of-the-art approaches, and incorporating sequence features further improves its prediction accuracy (*Figure 2—figure supplement 6*; *Figure 2—figure supplement 6—source data 1* and *Figure 2—figure supplement 6—source data 2*). We also performed 10-fold cross-validation on the binding affinities of protein-DNA pairs in this dataset and found that IDEA outperforms a recent regression model that considers the shape of DNA with different sequences (*Yang et al., 2017*; *Figure 2—figure supplement 7*). Details are provided in the Appendix 1 *section: Comparison of IDEA predictive performance using HT-SELEX data*.

We also applied IDEA to predict the binding affinity of additional TFs with available MITOMI measurements *Geertz et al., 2012*. For PHO4, another bHLH TF, IDEA's predictions, trained on the only available protein-DNA structure (PDB ID: 1A0A), correlate well with experimental measurements, showing a Pearson correlation coefficient of 0.60 and a Spearman's rank correlation coefficient of 0.63 (*Figure 2—figure supplement 8A and B*). We further evaluated IDEA's predictive performance

for the zinc-finger protein Zif268 (see the Materials and methods section *Structural modeling of protein and DNA*). Due to the limited experimental data for all possible DNA sequence combinations, we focused on testing IDEA's predictions on point-mutated DNA sequences. Predictions on point-mutated sequences are known to be a challenging task due to their minor deviations from the wild-type sequence. Despite this, IDEA achieves accurate predictions, with a Pearson correlation coefficient of 0.57 and a Spearman's rank correlation coefficient of 0.60 (*Figure 2—figure supplement 8C and D*). We further expanded the training dataset to include all available Zif268 structures and their associated sequences from the same CATH zinc finger superfamily (CATH ID: 3.30.160.60). Incorporating these additional training data further improves the predictive accuracy (Pearson correlation coefficient 0.63; Spearman's rank correlation coefficient 0.60, *Figure 2—figure supplement 9*).

## IDEA demonstrates transferability across proteins in the same CATH superfamily

Since IDEA relies on the sequence-structure relationship of given protein-DNA complexes to reach predictive accuracy, we inquired whether the trained energy model from one protein-DNA complex could be generalized to predict the sequence-specific binding affinities of other complexes. To test this, we assessed the transferability of IDEA predictions across all 11 structurally available protein-DNA complexes within the MAX TF-associated CATH superfamily (CATH ID: 4.10.280.10, Helix-loop-helix DNA-binding domain). We trained IDEA based on each of these 11 complexes and then used the trained model to predict the MAX-based MITOMI binding affinity. Our results show that IDEA generally makes correct predictions of the binding affinity when trained on proteins that are homologous to MAX, with Pearson and Spearman correlation coefficients larger than 0.5 (*Figure 3A*, *Figure 3—figure supplement 1*).

The transferability of IDEA within the same CATH superfamily can be understood from the similarities in protein-DNA binding interfaces, which determine similar learned energy models. For example, the PHO4 protein (PDB ID: 1A0A) shares a highly similar DNA-binding interface with the MAX protein (PDB ID: 1HLO; *Figure 3B*), despite sharing only a 33.41% probability of being homologous. Consequently, the energy model derived from the PHO4-DNA complex (*Figure 3C*) exhibits a similar amino-acid-nucleotide interactive pattern as that learned from the MAX-DNA complex (*Figure 2B*). To further evaluate the similarity between the learned energy models and their connection to protein families, we performed principal component analysis (PCA) on the normalized energy models across 24 proteins from 12 protein families (*Yang et al., 2017*). Our analysis (*Figure 3—figure supplement 2*) reveals that most of the energy models from the same protein family fall within the same cluster, while those from different protein families exhibit distinct patterns. Moreover, the relative distance between energy models in PCA space reflects the degree of transferability between them. For example, PHO4 (PDB ID: 1A0A) is positioned close to MAX (PDB ID: 1HLO), whereas USF1 (PDB ID: 1AN4) and TCF4 (PDB ID: 6OD3) are farther away. This is consistent with the results in *Figure 3A*, where the energy model trained on PHO4 has better transferability than those trained on USF1 or TCF4.

## Identification of genomic protein-DNA binding sites

The genomic locations of DNA binding sites are causally related to major cellular processes (*Furey, 2012*). Although multiple techniques have been developed to enable HT mapping of genomic protein-DNA binding locations, such as ChIP-seq (*Park, 2009*; *Furey, 2012*; *de Souza, 2012*), DAP-seq (*Bartlett et al., 2017*), and FAIRE-seq (*Giresi et al., 2007*), challenges remain for precisely pinpointing protein-binding sites at a base-pair resolution. Furthermore, the applicability of these techniques for different DNA-binding proteins is restricted by the quality of antibody designs (*Furey, 2012*). Therefore, a predictive computational framework would significantly reduce the costs and accelerate the identification of genomic binding sites of DNA-binding proteins.

We incorporated IDEA into a protocol to predict the genomic protein-DNA binding sites at the base-pair resolution, given a DNA-binding protein and a genomic DNA sequence. To evaluate the predictive accuracy of this protocol, we utilized publicly available ChIP-seq data for MAX TF binding in GM12878 lymphoblastoid cell lines (*Zhang et al., 2020*). As the experimental measurements were conducted at a 420 base pairs resolution, we averaged our modeling prediction over a window spanning 500 base pairs. Since the experimental signals are sparsely distributed across the genome, we focused our prediction on the 1 Mb region of Chromosome 1, which has the densest and most reliable

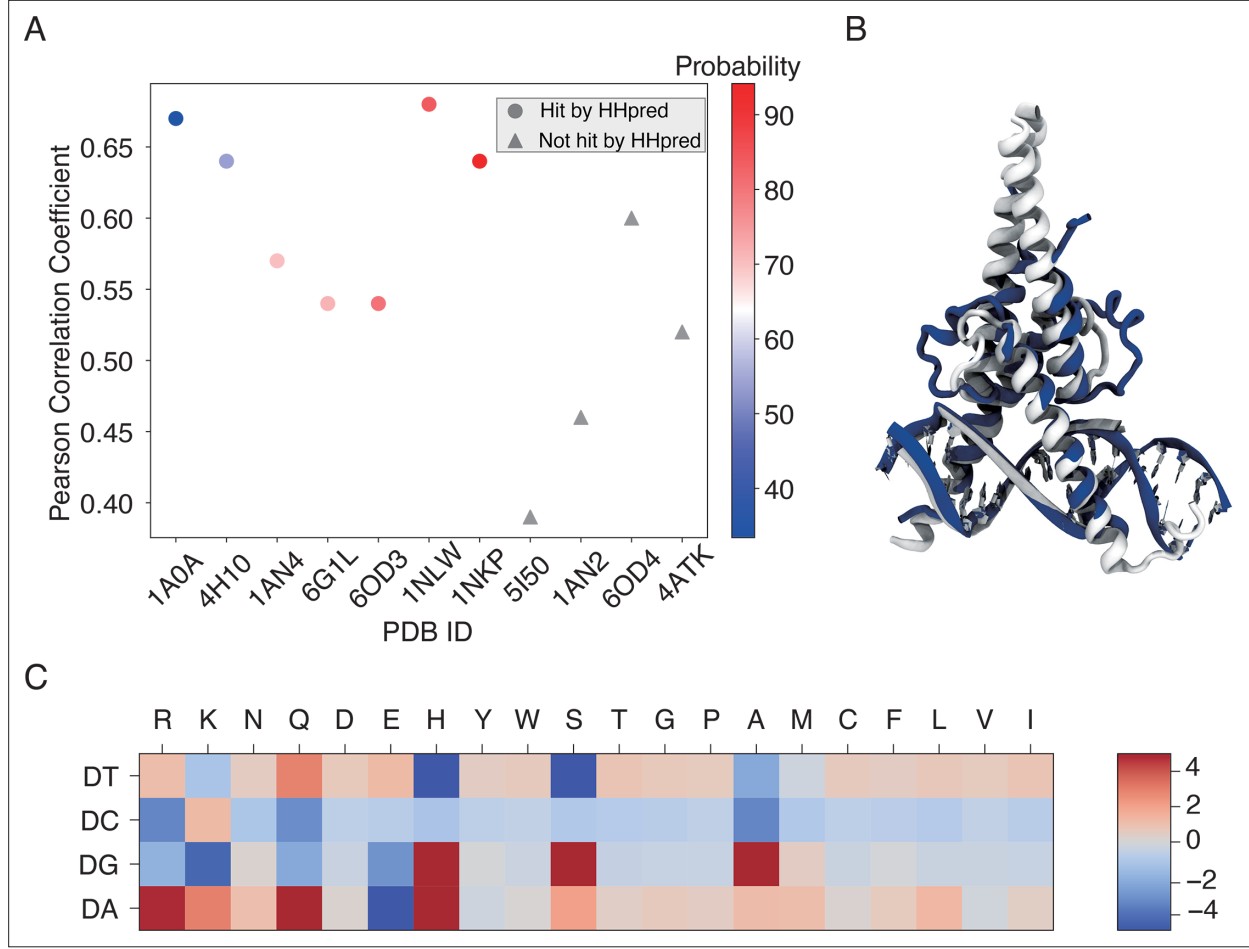

**Figure 3.** IDEA prediction shows transferability within the same CATH superfamily. (**A**) The predicted MAX binding affinity, trained on other protein-DNA complexes within the same protein CATH superfamily, correlates well with experimental measurement. The proteins are ordered by their probability of being homologous to the MAX protein, determined using HHpred (*Söding et al., 2005*). Training with a homologous protein (determined as a hit by HHpred) usually leads to better predictive performance (Pearson correlation coefficient > 0.5) compared to non-homologous proteins. (**B**) Structural alignment between 1HLO (white) and 1A0A (blue), two protein-DNA complexes within the same CATH Helix-loop-helix superfamily. The alignment was performed based on the E-box region of the DNA (*Humphrey et al., 1996*). (**C**) The optimized energy model for 1A0A, a protein-DNA complex structure of the transcription factor PHO4 and DNA, with 33.41% probability of being homologous to the MAX protein. The optimized energy model is presented in reduced units, as explained in the Materials and methods section: *Training protocol*.

The online version of this article includes the following figure supplement(s) for figure 3:

**Figure supplement 1.** Spearman's rank correlation coefficients between predicted and experimental MAX binding affinities across training proteins ordered by probability of being homologous to the MAX protein.

**Figure supplement 2.** Principal component analysis of the normalized IDEA-learned energy model across 12 protein families.

ChIP-seq signals (156,000 kb - 157,000 kb, representative window shown in *Figure 4A* top). In our model, lower predicted binding free energy corresponds to stronger protein-DNA binding affinity and thus a higher probability of a binding site. These predicted binding free energies were further normalized against those calculated from the randomized decoy sequences (see the Materials and methods section *Processing of ChIP-seq data* for details). The identified MAX-binding sites are marked in red, with a normalized Z score $< -0.75$. Our prediction successfully identifies the majority of the MAX binding sites, achieving an AUC score of 0.81 (*Figure 4B*). We also tested IDEA's predictive performance on two additional typical DNA-binding proteins and successfully identified their genomic binding sites (*Figure 4—figure supplement 1*). Therefore, IDEA provides an accurate initial prediction of genomic recognition sites of DNA-binding proteins based only on DNA sequences, facilitating more focused experimental efforts. Furthermore, IDEA holds the potential to identify binding sites beyond the experimental resolution and can aid in the interpretation of other sequencing techniques

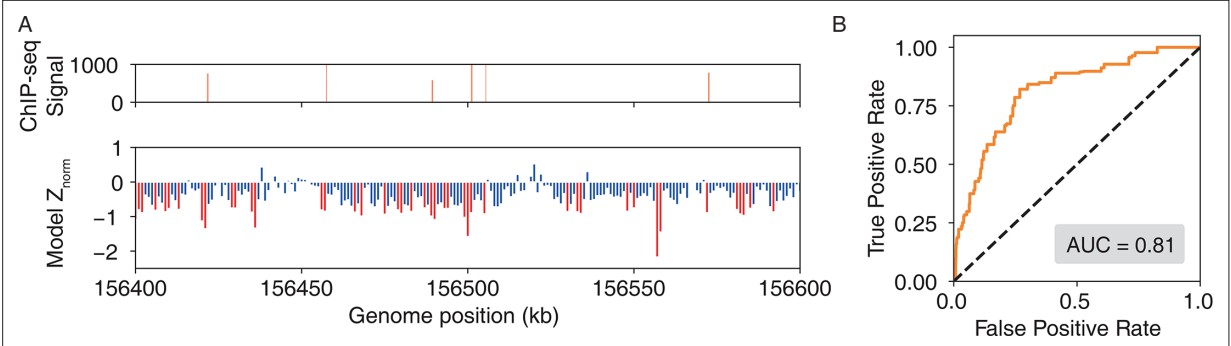

**Figure 4.** IDEA accurately identifies genome-wide protein-binding sites. (**A**) The IDEA-predicted MAX-transcription factor binding sites on the Lymphoblastoid cells chromosome (bottom) correlate well with ChIP-seq measurements (top), shown for a representative 1 Mb region of chromosome 1 where ChIP-seq signals are densest. For visualization purposes, a 1 kb resolution was used to plot the predicted normalized Z scores, with highly probable binding sites represented as red peaks. (**B**) AUC score for prediction accuracy based on the normalized Z scores, averaged over a 500 bp window to match the experimental resolution of 420 bp.

The online version of this article includes the following figure supplement(s) for figure 4:

**Figure supplement 1.** IDEA accurately identifies genomic binding sites for additional proteins.

to uncover additional factors that influence genome-wide DNA binding, such as chromatin accessibility (*Boyle et al., 2008*; *Corces et al., 2017*; *Granja et al., 2021*).

## Integration of IDEA into a residue-resolution simulation model captures protein-DNA binding dynamics

The trained residue-level model can be incorporated into a simulation model, thus enabling investigations into dynamic interactions between protein and DNA for mechanistic studies. Coarse-grained protein-nucleic acid models have shown strong advantages in investigating large biomolecular systems (*Savelyev and Papoian, 2010*; *Davtyan et al., 2012*; *Bascom and Schlick, 2018*; *Tan and Takada, 2020*; *Chakraborty et al., 2024*). Notably, the structure-based protein model (*Clementi et al., 2000*) and 3-Site-Per-Nucleotide (3SPN) DNA model *Freeman et al., 2014a* have been developed at the residue resolution to enable efficient sampling of biomolecular systems involving protein and DNA. A combination of these two models has proved instrumental in quantitatively studying multiple large chromatin systems (*Lin et al., 2021c*), including nucleosomal DNA unwrapping (*Zhang et al., 2016*; *Lequieu et al., 2016*; *Parsons and Zhang, 2019*), interactions between nucleosomes (*Moller et al., 2019*; *Lin and Zhang, 2024*), large chromatin organizations (*Ding et al., 2021*; *Liu et al., 2022*; *Lin and Zhang, 2024*), and its modulation by chromatin factors (*Watanabe et al., 2018*; *Leicher et al., 2020*; *Lin et al., 2021b*). Although both the protein and DNA force fields have been systematically optimized (*Noel et al., 2016*; *Knotts et al., 2007*; *Hinckley et al., 2013*; *Freeman et al., 2014a*; *Freeman et al., 2014b*), the non-bonded interactions between protein and DNA were primarily guided by Debye-Hückel treatment of electrostatic interactions, which did not consider sequence-specific interactions between individual protein amino acids and DNA nucleotides.

To refine this simulation model by including protein-DNA interaction specificity, we incorporated the IDEA-optimized protein-DNA energy model into the coarse-grained simulation model as a short-range interaction in the form of a Tanh function (see Materials and methods section *Sequence specific protein-DNA simulation model* for modeling details). This refined simulation model is used to simulate the dynamic interactions between DNA-binding proteins and their target DNA sequences. As a demonstration of the working principle of this modeling approach, we selected nine TF-DNA complexes whose binding affinities were measured by the same experimental group (*Privalov et al., 2011*). We applied IDEA to train an energy model based on these protein-DNA complexes, which yielded a modest correlation between the modeling-predicted binding affinity and experimental measurements (Pearson correlation coefficient 0.46, Spearman's rank correlation coefficient 0.55, *Figure 5A*). However, upon incorporating the learned energy model into our simulation model and computing protein-DNA binding free energies through umbrella sampling (*Torrie and Valleau, 1977*; *Figure 5B*), we observed a significant improvement in predictive accuracy. The predicted binding free energies

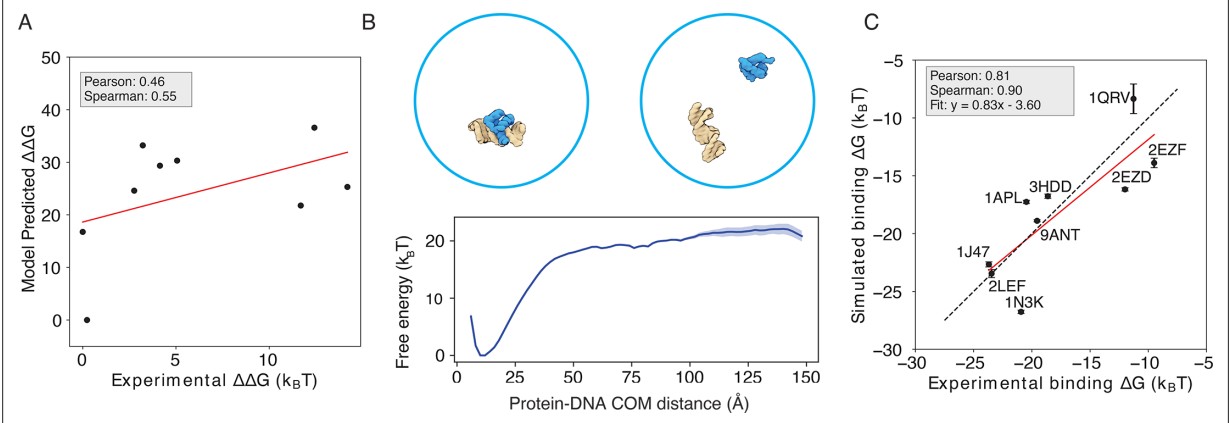

**Figure 5.** Enhanced protein-DNA simulation model by incorporating IDEA-optimized energy model. (**A**) Predicted protein-DNA binding free energies for 9 protein-DNA complexes using the IDEA-learned energy model correlate with the experimental measurements (*Privalov et al., 2011*). $\Delta\Delta G$ represents the changes in binding free energy relative to the protein-DNA complex with the lowest predicted value. The predicted binding free energies are presented in reduced units, as explained in *Materials and methods section Training protocol*. (**B**) Example of a protein-DNA complex structure (PDB ID: 9ANT) and the coarse-grained simulation used to evaluate protein-DNA binding free energy. A representative free energy profile was extracted from the simulation, using the center of mass (COM) distance between protein and DNA as the collective variable. The shaded region represents the standard deviation of the mean. Representative bound and unbound structures are shown above, with protein in blue and DNA in yellow. (**C**) Incorporating the IDEA-optimized energy model into the simulation model improves the prediction of protein-DNA binding affinity, compared to the prediction by the previous model with electrostatic interactions and uniform nonspecific attraction between protein and DNA (*Figure 5—figure supplement 2*). The predicted binding free energies are presented in physical units. Error bars represent the standard deviation of the mean from three equally partitioned segments of the simulation trajectory.

The online version of this article includes the following figure supplement(s) for figure 5:

**Figure supplement 1.** Binding free energy curves calculated from simulations based on non-sequence-specific homogenous electrostatic potential and IDEA potential models.

**Figure supplement 2.** Prediction of protein-DNA binding free energy with the non-sequence-specific homogeneous electrostatic potential model.

(*Kumar et al., 1992*; *Figure 5—figure supplement 1*) from our simulations correlate strongly with experimental measurements (Pearson correlation coefficient 0.81; Spearman's rank correlation coefficient 0.90) and near-perfectly reproduce the experimental binding affinities (*Figure 5C*). This prediction also represents a substantial improvement over the previous model, which depicts protein-DNA interactions through non-sequence-specific homogeneous electrostatic interactions (Pearson correlation coefficient 0.60, Spearman's rank correlation coefficient 0.70; *Figure 5—figure supplement 2*). Notably, due to an undetermined prefactor in the IDEA training, it can only provide relative binding free energies in reduced units for different protein-DNA complexes. In contrast, the IDEA-incorporated simulation model can predict the absolute binding free energies with physical units.

## Discussion

The protein-DNA interaction landscape has evolved to facilitate precise targeting of proteins towards their functional binding sites, which underlie essential processes in controlling gene expression. These interaction specifics are determined by physicochemical interactions between amino acids and nucleotides. By integrating sequences and structural data from available protein-DNA complexes into an interaction matrix, we introduce IDEA, a data-driven method that optimizes a system-specific energy model. This model enables HT in silico predictions of protein-DNA binding affinities and can be scaled up to predict genomic binding sites of DNA-binding proteins, such as TFs. IDEA achieves accurate de novo predictions using only protein-DNA complex structures and their associated sequences, but its accuracy can be further enhanced by incorporating available experimental data from other binding assay measurements, such as the SELEX data (*Jolma et al., 2010*; *Ogawa and Biggin, 2012*; *Isakova et al., 2017*), achieving accuracy comparable or better than state-of-the-art methods (*Figure 2—figure supplements 2 and 7*, *Figure 2—figure supplement 6—source data 1* and *Figure 2—figure supplement 6—source data 2*). Despite significant progress in genome-wide sequencing techniques

(*Giresi et al., 2007*; *Park, 2009*; *Furey, 2012*; *Bartlett et al., 2017*), determining sequence-specific binding affinities of DNA-binding biomolecules remains time-consuming and expensive. Therefore, IDEA presents a cost-effective alternative for generating the initial predictions before pursuing further experimental refinement.

A key advantage of IDEA is its incorporation of both structural and sequence information, which greatly reduces the demand for extensive training data. Prior efforts have applied deep learning techniques to predict the interactions between proteins and DNA based on their sequences (*Alipanahi et al., 2015*; *Zhou et al., 2015*; *Rastogi et al., 2018*). These approaches usually require a large amount of sequencing training data. By leveraging the sequence-structure relationship inherent in protein-DNA complex structures, IDEA achieves accurate de novo prediction using only a few protein-DNA complexes within the same protein superfamily.

In addition, despite the complicated in vivo environment, IDEA enables predictions of genomic binding sites and shows good agreement with the ChIP-seq measurements (*Figures 4*). Moreover, IDEA features rapid prediction, typically requiring 1–3 days to predict base-pair resolution genomic binding sites of a 1 Mb DNA region using a single CPU. Furthermore, IDEA's prediction can be trivially parallelized, making it possible to perform genome-wide predictions of DNA recognition sites by a given protein within weeks.

Another highlight of IDEA is its ability to present an interpretable, family-specific amino acid-nucleotide interaction energy model for given protein-DNA complexes. The optimized IDEA energy model can not only predict sequence-specific binding affinities of protein-DNA pairs but also provide a residue-specific interaction matrix that dictates the preferences of amino acid-nucleotide interactions within specific protein families (*Figure 3—figure supplement 2*). This interpretable energy matrix would facilitate the discovery of sequence binding motifs for target DNA-binding proteins, complementing both sequence-based (*Mukherjee et al., 2004*; *Aizenshtein-Gazit and Orenstein, 2022*; *Liu et al., 2025*) and structure-based approaches (*Wetzel et al., 2022*; *Chiu et al., 2023*; *Mitra et al., 2024*; *Oriol et al., 2024*). Additionally, we integrated this physicochemical-based energy model into a simulation framework, thereby improving the characterization of protein-DNA binding dynamics. IDEA-based simulation enables the investigation into dynamic interactions between various proteins and DNA, facilitating molecular-level understanding of the physical mechanisms underlying many DNA-binding processes, such as transcription, epigenetic regulations, and their modulation by sequence variations, such as single-nucleotide polymorphisms (SNPs; *Hindorff et al., 2009*; *Lappalainen et al., 2019*).

Since the IDEA-optimized energy model serves as a 'molecular grammar' for guiding protein-DNA interaction specifics, it can be expanded to include additional variants of amino acids and nucleotides. With recent advancements in sequencing and structural characterization (*Metzker, 2010*; *Doerr, 2017*), as well as deep-learning-based structural predictions (*Abramson et al., 2024*; *Tunyasuvunakool et al., 2021*; *Baek et al., 2024*), thousands of structures and sequences have been solved for these variants. The IDEA procedure can be repeated for these variant-included structures to expand the optimized energy model by including modifications such as methylated DNA nucleotides (*Moore et al., 2013*) and post-translationally modified amino acids (*Strahl and Allis, 2000*). Such strategies will facilitate studies on the functional relevance of epigenetic variants, such as those caused by exposure to environmental hazards (*Jirtle and Skinner, 2007*), and their structural impact on the human genome (*Portela and Esteller, 2010*; *Allis and Jenuwein, 2016*; *de Goede et al., 2021*).

Although IDEA has proven successful in many examples, it can be improved in several aspects. The model currently assumes the training and testing sequences share the same protein-DNA structure. While double-stranded DNA is generally rigid, recent studies have shown that sequence-dependent DNA shape contributes to their binding specificity (*Zhou et al., 2015*; *Li et al., 2024*; *Mitra et al., 2024*). To improve predictive accuracy, one could incorporate predicted DNA shapes or structures into the IDEA training protocol. In addition, the model is residue-based and evaluates the binding free energy as the additive sum of contributions from individual amino-acid-nucleotide contacts. This assumption does not account for cooperative effects that may arise from multiple nucleotide changes. A potential refinement could utilize a finer-grained model that includes more atom types within contacting residues and employs a many-body potential to account for such cooperative effects.

# Materials and methods

## Structural modeling of protein and DNA

We utilized a coarse-grained framework to extract structural information of protein and DNA at the residue level (*Figure 1*). Specifically, each protein amino acid was represented by the coordinates of the Cα atom, and each DNA nucleotide was represented by either the P atom in the phosphate group or the C5 atom on the nucleobase. The choice between these two DNA atom types (C5 or P) depended on their distances to surrounding Cα atoms. We hypothesized that DNA atoms closer to Cα atoms would provide more meaningful information on protein-DNA interactions. Therefore, we chose the DNA atom type (C5 or P) with the largest sum of occurrences within 8 Å, 9 Å, and 10 Å distances from their surrounding Cα atoms (*Appendix 1—table 1*). This selection rule was applied consistently across all complexes in the training dataset to maintain uniform criteria.

## Energy model optimization

### Training protocol

IDEA is a residue-based, data-driven biophysical model that predicts sequence-specific protein-DNA binding affinities. The model outputs relative binding free energies between protein and DNA, which can be converted to the relative binding affinity (represented by the dissociation constant $K_d$) using (*Stormo and Zhao, 2010*):

$$\Delta\Delta G_{\text{binding}} = RTln\Delta K_d \tag{1}$$

The model integrates both the sequence and structural information from experimentally determined protein-DNA structural interface into an interaction matrix for model training. These sequences, located at the protein-DNA structural interface (defined as those residues whose representative atoms have a distance $\leq 8$ Å, highlighted in blue in *Figure 1* Left), are hypothesized to have evolved to favor strong amino acid-nucleotide interactions. We collected the sequences from both the strong binders and decoy binders (considered weak binders) for model parameterization. Synthetic decoy binders were generated by randomizing either the DNA (1000 sequences) or protein sequences (10,000 sequences) of the strong binders (see *Appendix 1—figure 1* for an assessment of the model's robustness in transferability and generalizability across different numbers of decoys).

IDEA optimizes an amino-acid-nucleotide energy matrix, $\gamma(a, n)$, by maximizing the gap in binding energies between the strong and decoy binders. Similar strategies have been successfully applied in protein folding and protein-protein binding (*Bryngelson and Wolynes, 1987*; *Davtyan et al., 2012*; *Schafer et al., 2014*; *Lin et al., 2021a*; *Wang et al., 2024*). Specifically, the solvent-averaged binding free energies for strong binders ($E_{\text{strong}}$) and their corresponding decoy weak binders ($E_{\text{decoy}}$) are calculated using the following expression:

$$E_{\text{binding}} = \sum_{i\in\text{protein},j\in\text{DNA}} \gamma(a, n)\Theta_{i,j}(a_i, n_j) \tag{2}$$

Here, $\gamma(a, n)$ is a 20-by-4 residue-type-specific energy matrix encoding interaction energies between amino acid ($a$) and nucleotide ($n$). $\Theta_{i,j}(a_i, n_j)$ utilizes a switching function that captures an effective interaction range between each amino acid-nucleotide pair within the protein-DNA complex:

$$\Theta_{i,j}(a_i, n_j) = 0.5 \cdot \left(\tanh(\kappa \cdot (r_{i,j} - r_{\min})) \cdot \tanh(\kappa \cdot (r_{\max} - r_{i,j}))\right) + 0.5 \tag{3}$$

here, $r_{\min} = -8\,\mathring{A}$ and $r_{\max} = 8\,\mathring{A}$. This defines two residues to be "in contact" if their distance is less than 8 Å. $\kappa$ (here taken 0.7) is a scaling factor that modulates the steepness of the hyperbolic tangent function.

In practice, the $\gamma$ matrix was optimized to maximize $\delta E/\Delta E$, where $\delta E = \langle E_{\text{decoy}}\rangle - \langle E_{\text{strong}}\rangle$ represents the average energy gap between strong and decoy binders, and $\Delta E$ is the standard deviation of decoy binding energies. Mathematically,

$$\delta E = A(a, n)^T \gamma(a, n) \tag{4}$$

and

$$\Delta E = \sqrt{\gamma(a,n)^T B(a,n) \gamma(a,n)} \tag{5}$$

with

$$A(a,n) = \langle \phi_{\text{decoy}} \rangle - \langle \phi_{\text{strong}} \rangle \tag{6}$$

$$B(a,n) = \langle \phi_{\text{decoy}} \phi_{\text{decoy}}^T \rangle - \langle \phi_{\text{decoy}} \rangle \langle \phi_{\text{decoy}}^T \rangle \tag{7}$$

Here, $\phi(a,n) = \displaystyle\sum_{i \in \text{protein}, j \in \text{DNA}} \Theta_{i,j}(a_i, n_j)$ summarizes the total number of contacts between each type of amino acid and nucleotide in the given protein-DNA complexes. The vector $A(a,n)$ quantifies the difference in interaction frequency between the strong and decoy binders, with a dimension of 1×300. The covariance matrix $B(a,n)$ is a 300×300 matrix that quantifies the propensity of co-occurring residue type interactions in the decoy binders. Maximizing the function $\delta E / \Delta E$ with respect to $\gamma(a,n)$ corresponds to maximizing $A^T \gamma / \sqrt{\gamma^T B \gamma}$, which can be achieved by maximizing the functional objective $R(\gamma)$:

$$R(\gamma) = A^T \gamma - \lambda \sqrt{\gamma^T B \gamma} \tag{8}$$

where $\lambda$ is a Lagrange multiplier. Setting the derivative $\dfrac{\partial R(\gamma)}{\partial \gamma}$ equal to zero leads to $\gamma \propto B^{-1}A$. A filtering procedure is further applied to reduce the noise arising from the finite sampling of certain types of interactions: The matrix $B$ is first diagonalized such that $B^{-1} = P\Lambda^{-1}P^{-1}$, where $P$ is the matrix of eigenvectors of $B$, and $\Lambda$ is a diagonal matrix composed of $B$'s eigenvalues. We retained the principal $N$ modes of $B$, which are ordered by the descending magnitude of their eigenvalues, and replaced the remaining $300 - N$ eigenvalues in $\Lambda$ with the $N$th eigenvalue of $B$. In all optimizations reported in this work, $N = 70$ was generally a good choice to maximize the utilization of non-zero eigenvalues in the matrix $B$. For visualization purposes, the vector $\gamma$ is reshaped into a 20-by-4 matrix, as shown in *Figure 2B*.

To predict the binding free energy for a target protein-DNA pair, we substitute the target sequence into a known protein-DNA complex structure and recalculate the target $\phi$ matrix, which will be combined with the trained $\gamma$ matrix to compute the target protein-DNA binding free energies using *Equation 2*. Due to the presence of an undetermined scaling factor after the model optimization, the predicted free energies are presented in reduced units. Despite this, the relative free energies can be used to accurately rank binding affinities of target protein-DNA pairs. We utilize this computed solvent-averaged free energy to approximate the protein-DNA binding free energy in *Equation 1*. When the optimized energy is integrated into a simulation model, the IDEA-based simulations can predict the absolute binding free energies in physical units (*Figure 5C*).

## Treatment of complementary DNA sequences

To replace the DNA sequence in the protein-DNA complex structure with a target sequence, IDEA uses sequence identity to determine whether the target sequence belongs to the forward or reverse strand of the DNA in the protein-DNA structure. The more similar strand is selected and replaced with the target sequence. As the orientations of test sequences are specified from 5' to 3' in all datasets used in this study (e.g. processed MITOMI, HT-SELEX, and ChIP-seq data), this approach ensures that the target sequences are replaced and evaluated correctly. In cases where sequence orientation is not provided (though this was not an issue in this study), we recommend replacing both the forward and reverse strands with the target sequence separately and evaluating the corresponding protein-DNA binding free energies. Since strong binders are likely to dominate the experimental signals, the higher predicted binding affinity, with stronger binding free energies, should be taken as the model's final prediction.

## Enhanced modeling prediction with SELEX data

When additional experimental binding data, such as SELEX (*Rastogi et al., 2018*), is available, we extended IDEA to incorporate this data for enhancing the model's predictive capabilities. SELEX data provides binding affinities between given TFs and DNA sequences, which can be converted into dimensionless binding free energy using *Equation 1*, with $K_d$ being the normalized affinity generated from the SELEX package. We then calculated the $\phi$ value by utilizing the protein-DNA complex

structure, replacing the native DNA sequence with the SELEX-provided sequences. These $\phi$ values can be used together with the converted $\Delta\Delta G$ values to compute the $\gamma$ energy model based on *Equation 2*:

$$\gamma = \phi^{-1}\Delta\Delta G \tag{9}$$

here $\gamma$ is the enhanced protein-DNA energy model represented as a 300×1 vector. We estimated $\gamma$ by applying ridge regression (*Hoerl and Kennard, 1970*) to the linear system in *Equation 9*, using the previously computed $\phi$ matrix and the SELEX-derived $\Delta\Delta G$ values. A regularization parameter of $\alpha = 0.01$ was consistently applied throughout all computations to prevent overfitting.

For predicting the MAX-DNA binding specificity, we utilized the Human MAX SELEX data deposited in the European Nucleotide Archive database (accession number PRJEB25690) (*Rastogi et al., 2018*; *European Nucleotide Archive, 2024*). The R/Bioconductor package SELEX version 1.30.0 (*Rastogi et al., 2022*) was used to determine the observed R1 count for all 10-mers. Among them, ACCA CGTGGT is the motif with the highest frequency, which aligns with the DNA sequence in the MAX-DNA PDB structure (PDB ID: 1HLO), whose full DNA sequence is CACCACGTGGT. Consequently, we used this motif as the reference to align the remaining SELEX-measured 10-mer sequences. A total of 255 10-mer sequences, corresponding to the DNA variants measured from the MITOMI experiment (*Maerkl and Quake, 2007*), were selected. We threaded the PDB structure with these sequences to construct the $\phi$ and utilized the SELEX-converted binding free energies to compute the $\gamma$ energy model.

## Evaluation of IDEA predictions using HT-SELEX data

The processed HT-SELEX data used in this study are from *Yang et al., 2017*, which contains processed DNA binding sequence motifs and their normalized copy number detected from the HT-SELEX experiment, referred to as M-word scores. A higher M-word score corresponds to a stronger binding sequence motif detected in the HT-SELEX experiments. Among all the proteins in the data, 22 have experimentally determined protein-DNA complex structures. We predicted the binding free energies of all the processed sequences of these proteins using our protocol. For evaluating IDEA predictions, we classified motifs with M-word scores above 0.9 as strong binders (label 1) and those with M-word scores below 0.3 as weak binders (label 0). In cases where no sequences had an M-word score below 0.3, we used alternative cutoffs of 0.4 or 0.5 to ensure that at least three weak-binding sequences were included. These labeled strong and weak binders were then used as the ground truth for evaluating IDEA's predictions. We assessed IDEA's predictive accuracy by calculating the probability density distributions of the predicted binding free energies (*Figure 2—figure supplement 4*). A well-separated distribution between strong and weak binders indicates a successful prediction. Additionally, we calculated the AUC score and balanced PRAUC score based on the Receiver Operating Characteristic (ROC) and Precision-Recall (PR) analyses of these probability distributions. More details about the evaluation metrics and comparisons with popular existing models are provided in the Appendix 1 *section: Comparison of IDEA predictive performance using HT-SELEX data.*

## Selection of CATH superfamily for testing model transferability

The MAX protein is characterized by a Helix-loop-helix DNA-binding domain and belongs to the CATH superfamily 4.10.280.10 (*Sillitoe et al., 2021*), which includes a total of 11 protein-DNA structures with the E-box sequence (CACGTG), similar to the MAX-binding DNA. We hypothesized that proteins within the same superfamily exhibit common residue-interaction patterns due to their structural and evolutionary similarities. To test the transferability of our model, we conducted individual training on each one of those 11 protein-DNA complex structures and their associated sequences, predicting the MAX-DNA binding affinity in the MITOMI dataset (*Maerkl and Quake, 2007*). To examine the connection between the predictive outcome and the probability that the training protein is homologous to MAX, HHpred (*Gabler et al., 2020*) was utilized to search for MAX's homologous proteins, using the sequence of 1HLO as the query (*Figure 3A*).

## Processing of ChIP-seq data

To evaluate the predictive capabilities of our protocol on specific chromosomal segments, we selected ChIP-seq data for three TFs—MAX, EGR1, and CTCF. The data were sourced from the ENCODE

project, with accession numbers ENCFF361EVH (MAX), ENCSR026GSW (EGR1), and ENCFF960ZGP (CTCF).

For testing, we chose genomic regions with the densest ChIP-seq signals. For MAX and CTCF, we focused on the human GM12878 chromosome 1 (GRCh38) region between 156,000,000 bp and 157,000,000 bp, as this 1 Mb segment contains the densest signals for both TFs, with 33 MAX binding sites and 70 CTCF binding sites. For EGR1, we used the HepG2 cell line, focusing on the GRCh38 chr8 region between 143,000,000 bp and 144,000,000 bp, which contains 97 peaks.

To predict binding sites, we scanned the corresponding DNA regions base by base using the DNA sequence closest to each protein in the relevant protein-DNA complex structures. For MAX, we used the 11 bp sequence from the PDB structure 1HLO, generating $1,000,000 - 11 + 1 = 999,990$ sequences. For EGR1, we used the 10 bp sequence from PDB 1AAY, producing $1,000,000 - 10 + 1 = 999,991$ sequences. For CTCF, no single PDB structure covers the entire protein (*Yang et al., 2023*), so two PDB structures were used: 8SSS (ZF1-ZF7, 23 bp) and 8SSQ (ZF3–ZF11, 35 bp), generating $1,000,000 - 23 + 1 = 999,978$ and $1,000,000 - 35 + 1 = 999,966$ sequences, respectively.

Predicted binding free energies for each TF were normalized into Z-scores using randomized decoy sequences:

$$Z_{norm} = \frac{E - \langle E_{decoy} \rangle}{\text{SD}(E_{decoy})} \tag{10}$$

where $\langle \ \rangle$ is the average over all decoy sequences, and SD is the standard deviation. Given the evolutionary preference for strong binding, most Z-scores were negative, indicating stronger binding. ROC curves were then constructed to assess the predictive performance, with lower predicted Z-scores representing stronger binding affinity (*Figure 4—figure supplement 1*).

## Sequence-specific protein-DNA simulation model

We utilized a previously developed protein and DNA residue-level coarse-grained model for simulating protein-DNA binding interactions. The protein was modeled using the Cα-based structure-based model (*Clementi et al., 2000*), with each amino acid represented by a simulation bead located at the Cα site. The DNA was modeled with the 3-Site-Per-Nucleotide (3SPN) DNA model (*Freeman et al., 2014a*), with each DNA nucleotide modeled by three coarse-grained sites located at the centers of mass of the phosphate, sugar, and base sites. Both these two models and their combination have been validated to reproduce multiple experimental measurements (*Clementi et al., 2000*; *Hinckley et al., 2013*; *Freeman et al., 2014a*; *Lequieu et al., 2016*; *Lequieu et al., 2017*; *Moller et al., 2019*; *Leicher et al., 2020*; *Ding et al., 2021*; *Lin et al., 2021b*; *Liu et al., 2022*).

Detailed descriptions of this protein-DNA force field have been documented in previous studies (*Zhang et al., 2016*; *Ding et al., 2021*; *Lin et al., 2021b*; *Liu et al., 2022*). Specifically, we followed (*Freeman et al., 2014a*; *Mitra et al., 2025*) for simulating DNA-DNA interactions and (*Bulyk et al., 2001*; *Clementi et al., 2000*) for simulating the protein-protein interactions. The protein-DNA interactions were modeled with a long-range electrostatic interaction and a short-range sequence-specific interaction. The electrostatic interactions were simulated with the Debye-Hückel treatment to capture the screening effect at different ionic concentrations:

$$U_{elec}(r) = \frac{1}{4\pi\epsilon_0} \frac{q_i q_j}{\epsilon r} e^{-r/l_D} \tag{11}$$

where $\epsilon = 78.0$ is the dielectric constant of the bulk solvent. $q_i$ and $q_j$ correspond to the charges of the two particles. $\epsilon_0 = 1$ is the vacuum electric permittivity, and $l_D$ is the Debye-Hückel screening length.

The excluded volume effect was modeled using a hard wall potential with an apparent radius of σ = 4.0 Å:

$$U_{exclude}(r) = 4\epsilon(\frac{\sigma}{r})^{12} \tag{12}$$

An additional sequence-specific protein-DNA interaction was modeled with a Tanh function between the phosphate site of the DNA and the amino acid, with the form:

$$V_{\mathrm{PD}}(r) = \frac{W}{2}(1 + \tanh[\eta(r_0 - r)]) \tag{13}$$

where $r_0 = 8$ Å and $\eta = 0.7$ Å$^{-1}$ is a 4×20 matrix that characterizes the interaction energy between each type of amino acid and DNA nucleotide. Here, we used the normalized IDEA-learned energy model based on 9 protein-DNA complexes (*Figure 5A*). The IDEA-learned energy model was normalized by the sum of the absolute value of each term of the energy model. Detailed parameters are provided in *Appendix 1—table 2*. Additional details on the protein-DNA simulation are also provided in Appendix 1.

## Acknowledgements

This work was supported by the startup funding from North Carolina State University. Additionally, this research received partial funding from the NC State Genetics and Genomics Academy and Comparative Medicine Institute. We appreciate Dr. Keith Weninger, Dr. Faruck Morcos, and Dr. Qin Zhou for their insightful discussions and Dr. Sebastian Maerkl for guiding us to the MITOMI experimental data.

## Additional information

### Funding

| Funder | Grant reference number | Author |
|---|---|---|
| North Carolina State University | Startup funding | Yafan Zhang Irene Silvernail Zhuyang Lin Xingcheng Lin |
| North Carolina State University | Interdisciplinary seed funding | Yafan Zhang Irene Silvernail Zhuyang Lin Xingcheng Lin |
| North Carolina State University | Ideation fund | Yafan Zhang Irene Silvernail Zhuyang Lin Xingcheng Lin |

The funders had no role in study design, data collection and interpretation, or the decision to submit the work for publication.

### Author contributions

Yafan Zhang, Data curation, Software, Formal analysis, Validation, Investigation, Visualization, Methodology, Writing – original draft, Writing – review and editing; Irene Silvernail, Data curation, Software, Formal analysis, Validation, Investigation, Methodology, Writing – original draft, Writing – review and editing; Zhuyang Lin, Software, Investigation, Methodology; Xingcheng Lin, Conceptualization, Resources, Data curation, Software, Formal analysis, Supervision, Funding acquisition, Validation, Investigation, Visualization, Methodology, Writing – original draft, Project administration, Writing – review and editing

### Author ORCIDs

Yafan Zhang ⓘ https://orcid.org/0000-0002-7867-2873
Irene Silvernail ⓘ https://orcid.org/0009-0003-3070-974X
Zhuyang Lin ⓘ https://orcid.org/0009-0009-0480-7024
Xingcheng Lin ⓘ https://orcid.org/0000-0002-9378-6174

Reviewer #2 (Public review): https://doi.org/10.7554/eLife.105565.3.sa1
Reviewer #3 (Public review): https://doi.org/10.7554/eLife.105565.3.sa2
Author response https://doi.org/10.7554/eLife.105565.3.sa3

## Additional files

**Supplementary files**
MDAR checklist

**Data availability**
The MAX-DNA binding data were obtained from https://lbnc.epfl.ch/media/data/MAX_summary_ddGs.xls. The Pho4-DNA binding data were obtained from https://lbnc.epfl.ch/media/data/Pho4_Cbf1_summary_ddGs.xls. The Zif268-DNA binding data were obtained from https://www.pnas.org/doi/suppl/10.1073/pnas.1206011109/suppl_file/appendix.pdf. The SELEX data used in this paper were downloaded from https://www.ebi.ac.uk/ena/data/view/PRJEB25690 and https://rohslab.usc.edu/MSB2017/data-files.tar.gz. The ChIP-seq data used in this paper were obtained from the ENCODE Project: https://www.encodeproject.org/files/ENCFF361EVH/, https://www.encodeproject.org/files/ENCFF603PUS/, and https://www.encodeproject.org/files/ENCFF710VEH/. The implementation of the IDEA model, along with training and prediction examples, is available on our GitHub, copy archived at *LinResearchGroup-NCSU, 2025*, where we have also uploaded the original data and the essential scripts required to reproduce the results presented in the manuscript.

The following previously published datasets were used:

| Author(s) | Year | Dataset title | Dataset URL | Database and Identifier |
|---|---|---|---|---|
| Maerkl SJ, Quake SR | 2007 | MAX-DNA binding data | https://lbnc.epfl.ch/media/data/MAX_summary_ddGs.xls | EPFL LBNC data repository, MAX_summary_ddGs.xls |
| Maerkl SJ, Quake SR | 2007 | Pho4-DNA binding data | https://lbnc.epfl.ch/media/data/Pho4_Cbf1_summary_ddGs.xls | EPFL LBNC data repository, Pho4_Cbf1_summary_ddGs.xls |
| Kagda M | 2012 | MAX ChIP-seq IDR thresholded peaks | https://www.encodeproject.org/files/ENCFF361EVH/ | ENCODE, ENCFF361EVH |
| Kagda M | 2020 | EGR1 ChIP-seq IDR thresholded peaks | https://www.encodeproject.org/files/ENCFF603PUS/ | ENCODE, ENCFF603PUS |
| Kagda M | 2011 | CTCF ChIP-seq optimal IDR thresholded peaks | https://www.encodeproject.org/files/ENCFF710VEH/ | ENCODE, ENCFF710VEH |
| Rastogi C, Rube HT, Kribelbauer JF, Crocker J, Loker RE, Martini GD, Laptenko O, Freed-Pastor WA, Prives C, Stern DL, Mann RS, Bussemaker HJ | 2018 | SELEX data reported in this paper | https://www.ebi.ac.uk/ena/data/view/PRJEB25690 | European Nucleotide Archive, PRJEB25690 |
| Orenstein Y, Jolma A, Yin Y, Taipale J, Shamir R, Rohs R, Yang L | 2017 | M-word scores | https://rohslab.usc.edu/MSB2017/data-files.tar.gz | Rohs Lab website, data-files.tar.gz |

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

## Appendix 1

### Details of molecular dynamics simulation

The simulations were carried out using the LAMMPS software (*Thompson et al., 2022*) with the implementation of the protein-DNA force field (*Appendix 1—table 2*). We performed umbrella-sampling simulations (*Torrie and Valleau, 1977*) for comprehensive sampling of the binding and unbinding events to compute the protein-DNA binding affinities. To evaluate the model's performance, we selected 9 protein-DNA complexes with binding affinity measured by the same experimental lab (*Dragan et al., 2003a*; *Dragan et al., 2003b*; *Dragan et al., 2006*; *Dragan et al., 2004*; *Privalov et al., 2011*). Initial simulation configurations were prepared based on the crystal structures of the protein-DNA complexes. The PDB IDs of these complexes are provided in *Figure 5—figure supplements 1 and 2*.

The simulations were conducted under the same experimental conditions with 100 mM monovalent ions. We used a spring constant of 0.02 kcal/mol/Å$^2$ based on the center of mass (COM) distances between the protein and DNA. The centers of the umbrella windows were placed on a grid ranging from 0.0 to 140.0 Å, with a grid size of 10.0 Å, and each umbrella simulation lasted for 22 million steps, with a time step of 2.0 fs. We simulated 6 copies of the simulation with different random seeds to ensure sampling convergence. When reweighting the simulation to compute the free energy (*Kumar et al., 1992*; *Figure 5—figure supplements 1 and 2*), we excluded the first 2 million steps of each simulation trajectory. To calculate the error bars from our simulation, we divided each simulation trajectory into three equal-length blocks and independently calculated the reweighted free energies from them. The error bar was then estimated as the standard deviation of the three free energy profiles.

### Comparison of IDEA predictive performance using HT-SELEX data

To benchmark the performance of IDEA against state-of-the-art protein-DNA predictive models, we evaluated its ability to recognize strong binders with the HT-SELEX datasets across 22 proteins from 12 families (*Yang et al., 2017*). Specifically, we compare IDEA with two widely used sequence-based models: ProBound (*Rube et al., 2022*) and DeepBind (*Alipanahi et al., 2015*). ProBound has demonstrated superior performance over many other predictive protein-DNA models, including JASPAR 2018 (*Khan et al., 2018*), HOCOMOCO (*Kulakovskiy et al., 2018*), Jolma et al., 2013, and DeepSELEX (*Asif and Orenstein, 2020*). To use ProBound, we retrieved the trained binding model for each protein from https://motifcentral.org/home and used the GitHub implementation of ProBoundTools to infer the binding scores between protein and target DNA sequences. Except for POU3F1, binding models are available for all proteins. Therefore, we excluded POU3F1 and evaluated the protein-DNA binding affinities for the remaining 21 proteins. To use DeepBind, sequence-specific binding affinities were predicted directly with its web server. The Area Under the Curve (AUC) and the Precision-Recall AUC (PRAUC) scores were used as metrics for comparison. An AUC score of 1.0 indicates a perfect separation between the strong- and weak-binder distributions, while an AUC score of 0.5 indicates no separation. Because there is a significant imbalance in the number of strong and weak binders from the experimental data (*Yang et al., 2017*), where the strong binders are far fewer than the weak binders, we reweighted the samples to achieve a balanced evaluation, using 0.5 as the baseline for randomized prediction (*Saito and Rehmsmeier, 2015*). As summarized in *Figure 2—figure supplement 6* and *Figure 2—figure supplement 6—source data 1* and *Figure 2—figure supplement 6—source data 2*, IDEA ranked second among the three predictive models. In order to assess the performance of IDEA when augmented with additional protein-DNA binding data, we augmented IDEA using randomly selected half of the HT-SELEX data (see the Materials and methods section *Enhanced modeling prediction with SELEX data*). The augmented IDEA model achieved the best performance among all the models.

In addition, we compared the performance of IDEA with both general and family-specific knowledge-based energy models. First, we incorporated a knowledge-based generic protein-DNA energy model (DBD-Hunter) learned from the protein-DNA database, reported by Skoinick and coworkers (*Gao and Skolnick, 2008*), into our prediction protocol. This model assigns interaction energies to different functional groups within each DNA nucleotide, including phosphate (PP), sugar (SU), pyrimidine (PY), and imidazole (IM) groups. For our comparison, we averaged the energy contributions of these groups within each nucleotide and replaced the IDEA-learned energy model with the DBD-Hunter model to assess its ability to differentiate strong binders from weak binders in the HT-SELEX dataset (*Yang et al., 2017*). Additionally, we compared IDEA with rCLAMPS, a

family-specific energy model developed to predict protein-DNA binding specificity in the C2H2 and homeodomain families. rCLAMPS learns a position-dependent amino-acid-nucleotide interaction energy model. To incorporate this model into the binding free energy calculation, we averaged the energy contributions across all occurrences of each amino-acid-nucleotide pair, which resulted in a 20-by-4 residue-type-specific energy matrix. This matrix is structurally analogous to the IDEA-trained energy model and can be directly integrated into the binding free energy calculations. As shown in *Figure 2—figure supplement 6* and *Figure 2—figure supplement 6—source data 1* and *Figure 2—figure supplement 6—source data 2*, the IDEA model generally outperforms DBD-Hunter and rCLAMPS, demonstrating that it can achieve better predictive accuracy than both generic and family-specific knowledge-based models.

We also performed 10-fold cross-validation using the same HT-SELEX datasets, following the protocol described in the Materials and methods section *Enhanced modeling prediction with SELEX data*. For each protein, we divided the entire dataset into 10 equal, randomly assigned folds. In each iteration, we used randomly selected 9 of the 10 folds as the training dataset and the remaining fold as the testing dataset. This process was repeated 10 times so that each fold served as the test set once. We then reported the average $R^2$ scores across these iterations to evaluate IDEA's predictive performance. Our results are compared with the 1mer and 1mer +shape methods from *Yang et al., 2017*, the latest regression model that considers the shape of DNA with different sequences (*Figure 2—figure supplement 7*). This comparative analysis shows IDEA achieved higher predictive accuracy than the state-of-the-art sequence-based protein-DNA binding predictors for protein-DNA complexes that have available experimentally resolved structures.

Overall, these results demonstrate that IDEA can be used to predict the protein-DNA pairs in the absence of known binding sequence data, thus filling an important gap in protein-DNA predictions when experimental binding sequence data are unavailable.

## Analysis of examples where IDEA fails to recognize strong DNA binders

We examine IDEA's capability in identifying strong binders from the HT-SELEX dataset across 12 protein families (*Yang et al., 2017*). The model successfully predicts 18 out of 22 protein-DNA systems, but the performance is reduced in 4 cases. Closer investigations revealed the source of these limitations. In some instances, only the protein from a different organism is available. For example, the PDX1 HT-SELEX data utilized the human PDX1 protein, but no human PDX1-DNA complex structure is available. Therefore, the mouse PDX1-DNA complex structure (PDB ID: 2H1K) was used for model training. Differences between model organisms may reduce predictive accuracy. A similar limitation applies to POU3F1, where an available mouse complex (PDB ID: 4Y60) was used to predict human protein-DNA interactions. Notably, DeepBind (*Alipanahi et al., 2015*), a sequence-based prediction tool, also failed to distinguish strong from weak binders when using the mouse POU3F1 protein (AUC score: 0.457), but this was corrected with the human POU3F1 protein (AUC score: 0.956).

IDEA also fails to fully resolve the binding preference of USF1. A closer examination of the HT-SELEX data reveals a lack of distinction among the sequences, as most sequences, including those with the lowest M-word (binding affinity) scores, contain the DNA-binding E-box sequence CACGTG. Therefore, USF1 represents a challenging example where the experimental data only consists of strong binders with limited variations in binding affinity, which likely results from differences in flanking sequences of the E-box motif.

Egr1 stands as a peculiar example. Whereas IDEA does not effectively distinguish between the strong and weak binders in the current HT-SELEX dataset, its predictions are consistent with other experimental datasets, including binding affinities measured by k-MITOMI (*Geertz et al., 2012*; *Figure 2—figure supplement 8A and B*), preferred binding sequences from protein-binding microarray, an earlier HT-SELEX experiment, and bacterial one-hybrid data (*Stormo and Zhao, 2010*). Therefore, further investigation of the current HT-SELEX data is needed to reconcile these differences.

**Appendix 1—table 1.** This table reports the number of atoms within the cutoff distances from the Cα atoms for all the protein-DNA structures used in this study.

For each structure, we selected the DNA atom type (C5 or P) with the largest sum of occurrences within 8 Å, 9 Å, and 10 Å distances from surrounding Cα atoms for modeling. For protein YY1 (PDB ID: 1ubd), where the counts for C5 and P were identical, both models trained on C5 and P achieved an AUC score and PRAUC score of 1.0 for distinguishing strong from weak binders of the HT-SELEX dataset (*Figure 2—figure supplement 4*). The $R^2$ values for 10-fold cross-validation on the HT-SELEX dataset were 0.68 for the C5-trained model and 0.647 for the P-trained model. *Figure 2— figure supplement 4*; *Figure 2—figure supplement 5*; *Figure 2—figure supplement 7* use the result from the C5-trained model for YY1.

|  | 8 Å | 9 Å | 10 Å | Sum |
|---|---|---|---|---|
| 1a0a (C5) | 15 | 22 | 22 | 59 |
| 1a0a (P) | 18 | 19 | 20 | 57 |
| 1aay (C5) | 11 | 15 | 16 | 42 |
| 1aay (P) | 14 | 15 | 15 | 44 |
| 1an4 (C5) | 6 | 11 | 16 | 33 |
| 1an4 (P) | 16 | 18 | 24 | 58 |
| 1apl (C5) | 7 | 11 | 14 | 32 |
| 1apl (P) | 10 | 14 | 15 | 39 |
| 1dux (C5) | 8 | 9 | 11 | 28 |
| 1dux (P) | 6 | 7 | 12 | 25 |
| 1gu4 (C5) | 14 | 18 | 20 | 52 |
| 1gu4 (P) | 13 | 16 | 19 | 48 |
| 1hlo (C5) | 11 | 15 | 19 | 45 |
| 1hlo (P) | 13 | 14 | 15 | 42 |
| 1ig7 (C5) | 6 | 8 | 11 | 25 |
| 1ig7 (P) | 9 | 12 | 13 | 34 |
| 1j47 (C5) | 9 | 12 | 16 | 37 |
| 1j47 (P) | 16 | 18 | 18 | 52 |
| 1nk3 (C5) | 5 | 9 | 11 | 25 |
| 1nk3 (P) | 8 | 9 | 10 | 27 |
| 1nkp (C5) | 11 | 16 | 20 | 47 |
| 1nkp (P) | 16 | 20 | 20 | 56 |
| 1nlw (C5) | 11 | 16 | 20 | 47 |
| 1nlw (P) | 14 | 19 | 20 | 53 |
| 5cbx (C5) | 8 | 15 | 17 | 40 |
| 5cbx (P) | 14 | 14 | 15 | 43 |
| 6od3 (C5) | 5 | 8 | 10 | 23 |
| 6od3 (P) | 8 | 11 | 11 | 30 |
| 7tdw (C5) | 6 | 9 | 10 | 25 |
| 7tdw (P) | 8 | 10 | 11 | 29 |
| 7xv6 (C5) | 9 | 16 | 20 | 45 |
| 7xv6 (P) | 14 | 16 | 18 | 48 |

*Appendix 1—table 1 continued on next page*

*Appendix 1—table 1 continued*

|  | 8 Å | 9 Å | 10 Å | Sum |
|---|---|---|---|---|
| 8e3d (C5) | 13 | 20 | 23 | 56 |
| 8e3d (P) | 20 | 22 | 24 | 66 |
| 8k8d (C5) | 11 | 16 | 18 | 45 |
| 8k8d (P) | 9 | 14 | 16 | 39 |
| 8osb (C5) | 13 | 22 | 27 | 62 |
| 8osb (P) | 26 | 30 | 34 | 90 |
| 8pm7 (C5) | 4 | 8 | 10 | 22 |
| 8pm7 (P) | 9 | 13 | 13 | 35 |
| 8ssq (C5) | 28 | 41 | 46 | 115 |
| 8ssq (P) | 34 | 41 | 46 | 121 |
| 8sss (C5) | 29 | 39 | 42 | 110 |
| 8sss (P) | 28 | 35 | 35 | 98 |
| 9ant (C5) | 5 | 7 | 11 | 23 |
| 9ant (P) | 8 | 10 | 12 | 30 |
| 1ozj (C5) | 9 | 13 | 21 | 43 |
| 1ozj (P) | 13 | 18 | 20 | 51 |
| 1qrv (C5) | 5 | 9 | 13 | 27 |
| 1qrv (P) | 9 | 9 | 11 | 29 |
| 1ubd (C5) | 14 | 21 | 24 | 59 |
| 1ubd (P) | 17 | 19 | 23 | 59 |
| 2ezd (C5) | 6 | 9 | 11 | 26 |
| 2ezd (P) | 10 | 11 | 11 | 32 |
| 2ezf (C5) | 7 | 9 | 10 | 26 |
| 2ezf (P) | 6 | 7 | 10 | 23 |
| 2h1k (C5) | 6 | 8 | 10 | 24 |
| 2h1k (P) | 8 | 11 | 12 | 31 |
| 2lef (C5) | 12 | 18 | 21 | 51 |
| 2lef (P) | 20 | 20 | 21 | 61 |
| 2wty (C5) | 13 | 19 | 23 | 55 |
| 2wty (P) | 17 | 20 | 20 | 57 |
| 2xsd (C5) | 11 | 14 | 14 | 39 |
| 2xsd (P) | 10 | 11 | 12 | 33 |
| 3hdd (C5) | 5 | 7 | 10 | 22 |
| 3hdd (P) | 7 | 8 | 11 | 26 |
| 4hc7 (C5) | 5 | 10 | 12 | 27 |
| 4hc7 (P) | 8 | 11 | 13 | 32 |
| 4xrs (C5) | 5 | 11 | 17 | 33 |
| 4xrs (P) | 11 | 14 | 15 | 40 |
| 4y60 (C5) | 7 | 11 | 14 | 32 |

*Appendix 1—table 1 continued*

|  | 8 Å | 9 Å | 10 Å | Sum |
|---|---|---|---|---|
| 4y60 (P) | 15 | 15 | 17 | 47 |

**Appendix 1—table 2.** Summary of parameters used in our protein-DNA simulation model (unit: kcal/mol).

|  | PP | SU | A | G | C | T |
|---|---|---|---|---|---|---|
| ALA | 0 | 0 | –0.0228 | 0.0101 | 0.0199 | –0.0061 |
| ARG | 0 | 0 | –0.0402 | 0.034 | –0.0363 | –0.021 |
| ASN | 0 | 0 | –0.0163 | 0.0252 | 0.0159 | –0.003 |
| ASP | 0 | 0 | –0.0001 | 0.012 | 0.0048 | 0.0101 |
| CYS | 0 | 0 | 0.0025 | 0.0107 | 0.0062 | 0.0075 |
| GLU | 0 | 0 | 0.0045 | –0.021 | 0.0189 | –0.0059 |
| GLN | 0 | 0 | –0.0138 | –0.006 | 0.0094 | –0.0032 |
| GLY | 0 | 0 | 0.0027 | 0.032 | 0.0183 | –0.0486 |
| HIS | 0 | 0 | –0.003 | 0.0086 | 0.0149 | –0.0022 |
| ILE | 0 | 0 | –0.0188 | 0.0052 | 0.014 | 0.0099 |
| LEU | 0 | 0 | –0.0148 | –0.0216 | 0.0127 | 0.0121 |
| LYS | 0 | 0 | –0.0107 | 0.0083 | –0.0487 | 0.0115 |
| MET | 0 | 0 | –0.0049 | 0.0085 | 0.002 | 0.0143 |
| PHE | 0 | 0 | –0.0015 | –0.0117 | 0.0151 | –0.0042 |
| PRO | 0 | 0 | 0.0009 | 0.024 | –0.0073 | –0.0087 |
| SER | 0 | 0 | 0.012 | –0.01 | 0.0011 | –0.0079 |
| THR | 0 | 0 | –0.0213 | –0.0132 | 0.0163 | 0.0066 |
| TRP | 0 | 0 | –0.0181 | 0.0058 | 0.0172 | 0.003 |
| TYR | 0 | 0 | –0.0094 | 0.0016 | –0.0117 | 0.0127 |
| VAL | 0 | 0 | –0.006 | 0.0102 | 0.0013 | 0.006 |

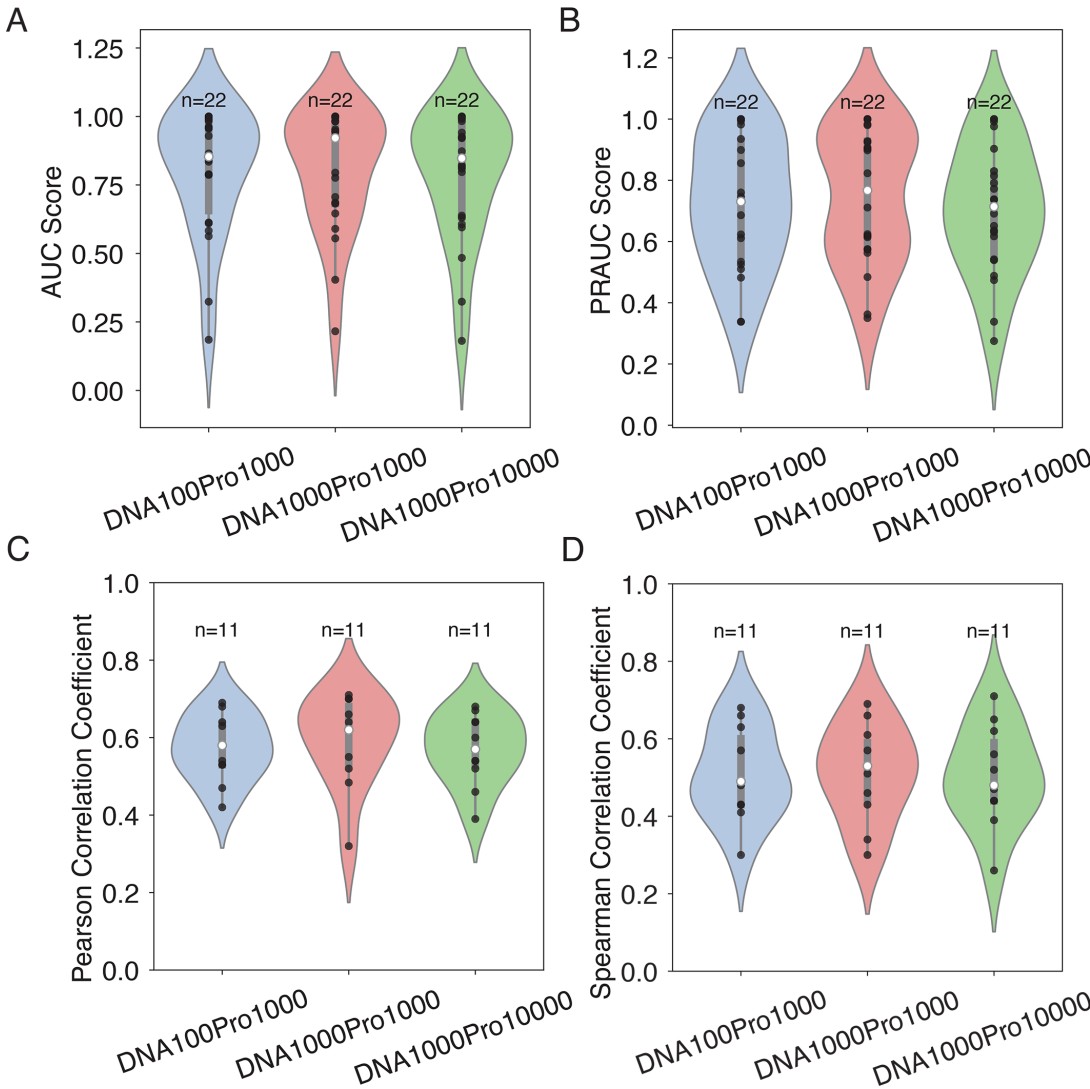

**Appendix 1—figure 1.** Effect of the number of decoy sequences on model generalizability and transferrability. (**A, B**) Evaluation of IDEA's generalizability across different decoy numbers, corresponding to *Figure 2E*. Violin plots summarize the distribution of AUC (**A**) and PRAUC (**B**) scores for three decoy combinations. (**C, D**) Analysis of IDEA transferability within the MAX CATH superfamily, corresponding to *Figure 3A*. Violin plots summarize the distribution of Pearson correlation coefficients (**C**) and Spearman's rank correlation coefficients (**D**) between predicted and experimental binding affinities for three decoy combinations. In each violin plot, the thick grey bar represents the interquartile range (first to third quartiles), and the thin line extends to 1.5 times the interquartile range. Individual data are depicted as scattered black points, and the white dot represents the median. Sample sizes (**n**) are labeled above each group. The three tested decoy combinations include: 100 DNA +1000 protein decoys (DNA100Pro1000), 1000 DNA +1000 protein decoys (DNA1000Pro1000), and 1000 DNA +10,000 protein decoys (DNA1000Pro10000). The consistent performance across all decoy combinations demonstrates the robustness of the model's prediction with respect to the number of decoy sequences.

