## [Editor Report · eLife Assessment]

This **valuable** work presents an interpretable protein-DNA Energy Associative (IDEA) model for predicting binding sites and affinities of DNA-binding proteins. While the method is **convincing**, it requires some adaptation for application to different proteins. The IDEA method is available and can be potentially used for predicting genome-wide protein-DNA binding sites.

---

## [Referee Report · Reviewer #2 (Public review)]

Summary:

Zhang et al. present a methodology to model protein-DNA interactions via learning an optimizable energy model, taking into account a represetative bound structure for the system and binding data. The methodology is sound and interesting. They apply this model for predicting binding affinity data and binding sites in vivo.

Strengths:

The manuscript is well organized with good visualizations and is easy to follow. The methodology is discussed in detail. The IDEA energy model seems like an interesting way to study a protein-DNA system in the context of a given structure and binding data. The authors show that an IDEA model trained on one system can be transferred to other structurally similar systems. The authors show good performance in discriminating between binding-vs-decoy sequences for various systems, and binding affinity prediction. The authors also show evidence of the ability to predict genome-wide binding sites.

Weaknesses:

An energy-based model which needs to be optimized for specific systems is inherently an uncomfortable idea. Prediction of binding affinity is a well-studied domain and many competitors exist, some of which are well used. The usefulness of this method will be a test of time. The methodology is interpretable in a limited sense. The model is dependent on preserved interface geometry which might lead to suboptimal results for novel folds. The model predicts different output for reverse complement sequence (which in reality are the same as far as double helix is concerned). This is unintuitive.

Comments on revisions:

The authors have addressed my points regarding comparisons with existing methods, clarifying discussion terminologies and proper discussion of the existing literature. This resulted in a stronger manuscript with a clearer understanding of applicability.

---

## [Referee Report · Reviewer #3 (Public review)]

Summary:

Protein-DNA interactions and sequence readout represent a challenging and rapidly evolving field of study. Recognizing the complexity of this task, the authors have developed a compact and elegant model. They applied well-established approaches to address a difficult problem, effectively enhancing the information extracted from sparse contact maps by integrating an artificial decoy sequence set and available experimental data. This has resulted in a practical tool that can be adapted for use with other proteins.

Strengths:

The authors integrate sparse information with available experimental data to construct a model whose utility extends beyond the limited set of structures used for training.

A comprehensive methods section is included, ensuring reproducibility.

The authors provide a well-represented performance comparison between their model and other existing models.

Additionally, the authors have shared their model as a GitHub project, reflecting their commitment to research transparency.

Weaknesses:

The coarse-graining procedure is quite convoluted, but the authors provide reasoning for the proposed scheme. The authors acknowledge discrepancies between data-driven and simulation models.

---

## [Author Response]

The following is the authors’ response to the original reviews

**Reviewer #1:**
Comment 0: Summary: This work presents an Interpretable protein-DNA Energy Associative (IDEA) model for predicting binding sites and affinities of DNA-binding proteins. Experimental results demonstrate that such an energy model can predict DNA recognition sites and their binding strengths across various protein families and can capture the absolute protein-DNA binding free energies.

We appreciate the reviewer’s careful assessment of the paper, and we thank the reviewer for the insightful suggestions and comments.

Comment 1: Strengths: (1) The IDEA model integrates both structural and sequence information, although such an integration is not completely original. (2) The IDEA predictions seem to have agreement with experimental data such as ChIP-seq measurements.

We appreciate the reviewer’s positive comments on the strength of the paper.

Comment 2: Weaknesses: (1) The authors claim that the binding free energy calculated by IDEA, trained using one MAX-DNA complex, correlates well with experimentally measured MAX-DNA binding free energy (Figure 2) based on the reported Pearson Correlation of 0.67. However, the scatter plot in Figure 2A exhibits distinct clustering of the points and thus the linear fit to the data (red line) may not be ideal. As such. the use of the Pearson correlation coefficient that measures linear correlation between two sets of data may not be appropriate and may provide misleading results for non-linear relationships.

We thank the reviewer for the insightful comments and agree that a linear fit between our predictions and the experimental data may not be the best measure of performance. The primary utility of the IDEA model is to predict high-affinity DNA-binding sequences for a given DNA-binding protein by assessing the relative binding affinities across different DNA sequences. In this regard, the ranked order of predicted sequence binding affinities serves as a better metric for evaluating the success of this model. To evaluate this, we calculated both Spearman’s rank correlation coefficient, which does not rely on linear correlation, and the Pearson correlation coefficient between our predictions and the experimental results. As shown in Figure 2, our computation shows a Spearman’s rank correlation coefficient of 0.65 for the MAX-based predictions using one MAX-DNA complex (PDB ID: 1HLO), supporting the model’s capability to effectively distinguish strong from weak binders.

Although our model generally captures the relative binding affinities across different DNA sequences, its predictive accuracy diminishes for low-affinity sequences (Figure 2).

This could be due to two limitations of the current modeling framework: (1) The model is residue-based and estimates binding free energy as the additive sum of contributions from individual contacting amino-acid-nucleotide pairs. This assumption does not account for cooperative effects caused by simultaneous changes at multiple nucleotide positions. One potential direction to further improve the model would be to use a finergrained representation by incorporating more atom types within contacting residues, and to use a many-body potential to better capture cooperative effects from multiple mutations. (2) The model assumes that the target DNA adopts the same binding interface as in the reference crystal structure. However, sequence-dependent DNA shape has been shown to be important in determining protein-DNA binding affinity [1]. To address this limitation, a future direction is to use deep-learning-based methods to incorporate predicted DNA shape or protein-DNA complex structures based on their sequences [2, 3] into our model prediction.

To fully evaluate the predictive power of IDEA, we have included Spearman’s rank correlation coefficient for every correlation plot in this manuscript and have updated the relevant texts. Across all our analyses, the Spearman’s rank correlation coefficients reveal similar predictive performance as the Pearson correlation coefficients. Additionally, we have included in our discussion the current limitations of our model and potential directions for future improvement.

We have edited our Discussion Section to include a discussion on the limitations of the current model. Specifically, the added texts are:

“Although IDEA has proved successful in many examples, it can be improved in several aspects. The model currently assumes the training and testing sequences share the same protein-DNA structure. While double-stranded DNA is generally rigid, recent studies have shown that sequence-dependent DNA shape contributes to their binding specificity [1, 2, 4]. To improve predictive accuracy, one could incorporate predicted DNA shapes or structures into the IDEA training protocol. In addition, the model is residue-based and evaluates the binding free energy as the additive sum of contributions from individual amino-acid-nucleotide contacts. This assumption does not account for cooperative effects that may arise from multiple nucleotide changes. A potential refinement could utilize a finer-grained model that includes more atom types within contacting residues and employs a many-body potential to account for such cooperative effects.”

Comment 3: (2) In the same vein, the linear Pearson Correlation analysis performed in Figure 5A and the conclusion drawn may be misleading.

We thank the reviewer for the insightful comments. As noted in our response to the previous comment, we have added Spearman’s rank correlation coefficient in addition to the Pearson correlation coefficient to all correlation plots, including Figure 5A.

Comment 4: (3) The authors included the sequences of the protein and DNA residues that form close contacts in the structure in the training dataset, whereas a series of synthetic decoy sequences were generated by randomizing the contacting residues in both the protein and DNA sequences. In particular, synthetic decoy binders were generated by randomizing either the DNA (1000 sequences) or protein sequences (10,000 sequences) from the strong binders. However, the justification for such randomization and how it might impact the model’s generalizability and transferability remain unclear.

We thank the reviewer for the insightful comments. The number of randomizing sequences was chosen to strike a balance between sufficient sequence coverage and computational feasibility. Because proteins have more types of amino acids than four nucleotides in DNA, we utilized more protein decoy sequences than DNA decoys. To examine the robustness of our choice against different number of decoy sequences, we repeated the transferability analysis within the bHLH superfamily (Figure 3A) and the generalizability analysis across 12 protein families (Figure 2E) using two additional decoy sequence combinations: (1) 1000 DNA sequences and 1000 protein sequences; (2) 100 DNA sequences and 1000 protein sequences. As shown in Figure S15, we achieved similar results to those reported using the original decoy set, demonstrating the robustness of our model prediction against the variations in the number of decoys. We have included this figure as Figure S15.

Comment 5: (4) The authors performed Receiver Operating Characteristic (ROC) analysis and reported the Area Under the Curve (AUC) scores in order to quantitate the successful identification of the strong binders by IDEA. It would be beneficial to analyze the precision-recall (PR) curve and report the PRAUC metric which could be more robust.

We agree with the reviewer that more robust statistical metrics should be used to evaluate our model’s performance. We have included the PRAUC score as an additional evaluation metric of the model’s performance. Due to a significant imbalance in the number of strong and weak binders from the experimental data [5], where the experimentally identified strong binders are far fewer than the weak binders, we reweighted the sample to achieve a balanced evaluation [6], using 0.5 as the baseline for randomized prediction. As shown in Figure S5, IDEA achieves successful predictions in 18 out of 22 cases, demonstrating its predictive accuracy.

The updated PRAUC result has been included as Figure S5 in the manuscript. We have also included the detailed precision-recall curves for each case in Figure S4.

In addition, we have provided PRAUC scores for comparing the performance of IDEA with other models, and have summarized these results in Table S2.

**Reviewer #2:**
Comment 0: Summary: Zhang et al. present a methodology to model protein-DNA interactions via learning an optimizable energy model, taking into account a representative bound structure for the system and binding data. The methodology is sound and interesting. They apply this model for predicting binding affinity data and binding sites in vivo. However, the manuscript lacks discussion of/comparison with state-of-the-art and evidence of broad applicability. The interpretability aspect is weak, yet over-emphasized.

We appreciate the reviewer’s excellent summary of the paper, and we thank the reviewer for the insightful suggestions and comments.

Comment 1: Strengths: The manuscript is well organized with good visualizations and is easy to follow. The methodology is discussed in detail. The IDEA energy model seems like an interesting way to study a protein-DNA system in the context of a given structure and binding data. The authors show that an IDEA model trained on one system can be transferred to other structurally similar systems. The authors show good performance in discriminating between binding-vs-decoy sequences for various systems, and binding affinity prediction. The authors also show evidence of the ability to predict genome-wide binding sites.

We appreciate the reviewer’s strong assessment of the strengths of this paper. We have further refined our Methods Section to ensure all modeling details are clearly presented.

Comment 2: Weaknesses: An energy-based model that needs to be optimized for specific systems is inherently an uncomfortable idea. Is this kind of energy model superior to something like Rosetta-based energy models, which are generally applicable? Or is it superior to family-specific knowledge-based models? It is not clear.

We thank the reviewer for the insightful comments. The protein-DNA energy model facilitates the calculation of protein-DNA binding free energy based on protein-DNA structures and sequences. Because this model is optimized using the structure-sequence relationship of given protein-DNA complexes, it features specificity based on the conserved structural interface characteristic of each protein family. Because of that, its predictive accuracy depends on the degree of protein-DNA interface similarity between the training and target protein-DNA pairs, and is distinct from a general protein-DNA energy model, such as a Rosetta-based energy model. The model has some connections to the familyspecific energy model. As shown in Author response image 1, systems belonging to the same protein superfamily (MAX and PHO4) exhibit similar patterns in their learned energy models, in contrast to those from a different superfamily (PDX1).

**Author response image 1. sa3fig1:** Comparison of learned energy models for different protein-DNA complexes: MAX (A), PHO4 (B), and PDX1 (C). MAX and PHO4 are members of the Helixloop-helix (HLH) CATH protein superfamily (4.10.280.100), while PDX1 belongs to another Homeodomain-like CATH protein superfamily (1.10.10.60).

To compare our approach with both general and family-specific knowledge-based energy models, we conducted two studies. First, we incorporated a knowledge-based generic protein-DNA energy model (DBD-Hunter) learned from the protein-DNA database, reported by Skoinick and coworkers [7], into our prediction protocol. This model assigns interaction energies to different functional groups within each DNA nucleotide (e.g., phosphate (PP), sugar (SU), pyrimidine (PY), and imidazole (IM) groups). For our comparison, we averaged the energy contributions of these groups within each nucleotide and replaced the IDEA-learned energy model with this generic one to test its ability to differentiate strong binders from weak binders in the HT-SELEX dataset [5]. As shown in Figure S6, the IDEA model generally achieves better performance than the generic energy model.

Additionally, we compared IDEA with rCLAMPS, a family-specific energy model developed to predict protein-DNA binding specificity in the C2H2 and homeodomain families.

As shown in Table S1 and Table S2, IDEA also shows better performance than rCLAMPS in most cases across the C2H2 and homeodomain families, demonstrating that it has better predictive accuracy than both state-of-the-art family-specific and generic knowledgebased models.

We have included relevant texts in Appendix Section Comparison of IDEA predictive performance Using HT-SELEX data to clarify this point. The added texts are:

In addition, we compared the performance of IDEA with both general and family-specific knowledge-based energy models. First, we incorporated a knowledgebased generic protein-DNA energy model (DBD-Hunter) learned from the protein-DNA database, reported by Skoinick and coworkers [7], into our prediction protocol. This model assigns interaction energies to different functional groups within each DNA nucleotide, including phosphate (PP), sugar (SU), pyrimidine (PY), and imidazole (IM) groups. For our comparison, we averaged the energy contributions of these groups within each nucleotide and replaced the IDEA-learned energy model with the DBD-Hunter model to assess its ability to differentiate strong binders from weak binders in the HTSELEX dataset [5]. Additionally, we compared IDEA with rCLAMPS, a familyspecific energy model developed to predict protein-DNA binding specificity in the C2H2 and homeodomain families. rCLAMPS learns a position-dependent amino-acid-nucleotide interaction energy model. To incorporate this model into the binding free energy calculation, we averaged the energy contributions across all occurrences of each amino-acid-nucleotide pair, which resulted in a 20-by-4 residue-type-specific energy matrix. This matrix is structurally analogous to the IDEA-trained energy model and can be directly integrated into the binding free energy calculations. As shown in Figure S6, Table S1, and Table S2, the IDEA model generally outperforms DBD-Hunter and rCLAMPS, demonstrating that it can achieve better predictive accuracy than both generic and family-specific knowledge-based models.

Comment 3: Prediction of binding affinity is a well-studied domain and many competitors exist, some of which are well-used. However, no quantitative comparison to such methods is presented. To understand the scope of the presented method, IDEA, the authors should discuss/compare with such methods (e.g. PMID 35606422).

We thank the reviewer for the insightful comments. As detailed in our response to Comment 5, we previously misused the term “binding specificity”, and would like to clarify that our model is designed to predict protein-DNA binding affinity. To compare the performance of IDEA with state-of-the-art protein-DNA predictive models, we examined the predictive accuracies of two additional popular computational models: ProBound [8] and DeepBind [9]. ProBound has been shown to have a better performance than several earlier predictive protein-DNA models, including JASPAR 2018 [11], HOCOMOCO [12], Jolma et al. [13], and DeepSELEX [14]. To benchmark these models’ performance, we examine each method’s capability to identify strong binders with the HT-SELEX datasets covering 22 proteins from 12 protein families [5]. As suggested by Reviewer 1, we also calculated the PRAUC score, reweighted to account for data imbalance [6], as a complementary metric for evaluating the model performance.

As shown in Figure S6, Table S1, and Table S2, IDEA ranked second among the three predictive methods. It is important to note that both ProBound and DeepBind were trained on a curated version of the HT-SELEX data [13], which overlaps with the testing data [5]. Compared with them, IDEA was trained only on the given structural and sequence information from a single protein-DNA complex, thus independent of the testing data. In order to assess how IDEA performs when incorporating knowledge from HT-SELEX data, we augmented the training by randomly including half of the HT-SELEX data (see the Methods Section Enhanced Modeling Prediction with SELEX Data). The augmented IDEA model achieved the best performance among all the models. Overall, IDEA can be used to predict protein-DNA affinities in the absence of known binding sequence data, thereby filling a critical gap when such experimental datasets are unavailable.

Additionally, we have conducted a 10-fold cross-validation using the same HT-SELEX data [5] and found that IDEA outperformed a recent regression model that considers the shape of DNA with different sequences [5].

We have revised our text to include the comparison between IDEA and other predictive models. Specifically, we revised the text in Section: IDEA Generalizes across Various Protein Families.

The revised text reads:

“To examine IDEA’s predictive accuracy across different DNA-binding protein families, we applied it to calculate protein-DNA binding affinities using a comprehensive HT-SELEX dataset [5]. We focused on evaluating the capability of IDEA to distinguish strong binders from weak binders for each protein with an experimentally determined structure. We calculated the probability density distribution of the top and bottom binders identified in the SELEX experiment. A well-separated distribution indicates the successful identification of strong binders by IDEA (Figure 2D and S4). Receiver Operating Characteristic (ROC) analysis was performed to calculate the Area Under the Curve (AUC) and the precision-recall curve (PRAUC) scores for these predictions. Further details are provided in the Methods Section Evaluation of IDEA Prediction Using HT-SELEX Data. Our analysis shows that IDEA successfully differentiates strong from weak binders for 80% of the 22 proteins across 12 protein families, achieving AUC and balanced PRAUC scores greater than 0.5 (Figure 2D and S5). To benchmark IDEA’s performance against other leading methods, we compared its predictions with several popular models, including the sequence-based predictive models ProBound [8] and DeepBind [9], the familybased energy model rCLAMPS [10], and the knowledge-based energy model DBD-Hunter [7]. IDEA demonstrates performance comparable to these stateof-the-art approaches, and incorporating sequence features further improves its prediction accuracy (Figure S6, Table S1, and Table S2). We also performed 10-fold cross-validation on the binding affinities of protein–DNA pairs in this dataset and found that IDEA outperforms a recent regression model that considers the shape of DNA with different sequences [5] (Figure S7). Details are provided in Section: Comparison of IDEA predictive performance Using HT-SELEX data.”

We also added one section Comparison of IDEA predictive performance Using HT-SELEX data in the Appendix to fully explain the comparison between IDEA and other popular models. The added texts are:

“To benchmark the performance of IDEA against state-of-the-art protein-DNA predictive models, we evaluated its ability to recognize strong binders with the HT-SELEX datasets across 22 proteins from 12 families [5]. Specifically, we compare IDEA with two widely used sequence-based models: ProBound [8] and DeepBind [9]. ProBound has demonstrated superior performance over many other predictive protein-DNA models, including JASPAR 2018 [11], HOCOMOCO [12], Jolma et al. [13], and DeepSELEX [14]. To use ProBound, we retrieved the trained binding model for each protein from motifcentral.org and used the GitHub implementation of ProBoundTools to infer the binding scores between protein and target DNA sequences. Except for POU3F1, binding models are available for all proteins. Therefore, we excluded POU3F1 and evaluated the protein-DNA binding affinities for the remaining 21 proteins. To use DeepBind, sequence-specific binding affinities were predicted directly with its web server. The Area Under the Curve (AUC) and the Precision-Recall AUC (PRAUC) scores were used as metrics for comparison. An AUC score of 1.0 indicates a perfect separation between the strong- and weak-binder distributions, while an AUC score of 0.5 indicates no separation. Because there is a significant imbalance in the number of strong and weak binders from the experimental data [5], where the strong binders are far fewer than the weak binders, we reweighted the samples to achieve a balanced evaluation, using 0.5 as the baseline for randomized prediction [6]. As summarized in Figure S6, Table S1, and Table S2, IDEA ranked second among the three predictive models. In order to assess the performance of IDEA when augmented with additional protein-DNA binding data, we augmented IDEA using randomly selected half of the HT-SELEX data (see the Methods Section Enhanced Modeling Prediction with SELEX Data). The augmented IDEA model achieved the best performance among all the models.”

“We also performed 10-fold cross-validation using the same HT-SELEX datasets, following the protocol described in the Methods Section Enhanced Modeling Prediction with SELEX Data. For each protein, we divided the entire dataset into 10 equal, randomly assigned folds. In each iteration, we used randomly selected 9 of the 10 folds as the training dataset and the remaining fold as the testing dataset. This process was repeated 10 times so that each fold served as the test set once. We then reported the average R2 scores across these iterations to evaluate IDEA’s predictive performance. Our results are compared with the 1mer and 1mer+shape methods from [5], the latest regression model that considers the shape of DNA with different sequences (Figure S7). This comparative analysis shows IDEA achieved higher predictive accuracy than the state-of-the-art sequence-based protein-DNA binding predictors for proteinDNA complexes that have available experimentally resolved structures.”

“Overall, these results demonstrate that IDEA can be used to predict the proteinDNA pairs in the absence of known binding sequence data, thus filling an important gap in protein-DNA predictions when experimental binding sequence data are unavailable.”

Comment 4: The term “interpretable” has been used lavishly in the manuscript while providing little evidence on the matter. The only evidence shown is the family-specific residue-nucleotide interaction/energy matrix and speculations on how these values are biologically sensible. Recent works already present more biophysical, fine-grained, and sometimes family-independent interpretability (e.g. PMID 39103447, 36656856, 38352411, etc.). The authors should put into context the scope of the interpretability of IDEA among such works.

We thank the reviewer for the insightful comment and agree that “interpretability” should be discussed in a relevant context. In our work, interpretability refers to the familyspecific amino-acid-nucleotide interaction energies identified from the model training, which reveal interaction preferences within protein-DNA binding interfaces. As detailed in our response to Comment 6, we performed principal component analysis (PCA) on the learned energy models and observed clustering of learned energy models corresponding to protein families. Therefore, the IDEA-learned energy models can be used as a signature to capture the energetic preferences of amino-acid-nucleotide interactions within a given protein family. This preference can be used to infer preferred sequence binding motifs, similar to those identified by other computational tools [10, 4, 15, 16].

We have revised the text to clarify the “interpretability” as the family-specific aminoacid-nucleotide interactions that govern sequence-dependent protein-DNA binding, and to discuss IDEA’s interoperability within the context of recent works, including those suggested by the reviewers.

We have revised the text in Introduction. The new text reads:

“Here, we introduce the Interpretable protein-DNA Energy Associative (IDEA) model, a predictive model that learns protein-DNA physicochemical interactions by fusing available biophysical structures and their associated sequences into an optimized energy model (Figure 1). We show that the model can be used to accurately predict the sequence-specific DNA binding affinities of DNA-binding proteins and is transferrable across the same protein superfamily. Moreover, the model can be enhanced by incorporating experimental binding data and can be generalized to enable base-pair resolution predictions of genomic DNA-binding sites. Notably, IDEA learns a family-specific interaction matrix that quantifies energetic interactions between each amino acid and nucleotide, allowing for a direct interpretation of the “molecular grammar” governing sequence-specific protein-DNA binding affinities. This interpretable energy model is further integrated into a simulation framework, facilitating mechanistic studies of various biomolecular functions involving protein-DNA dynamics.”

We have revised the text in Results. The new text reads:

“IDEA is a coarse-grained biophysical model at the residue resolution for investigating protein-DNA binding interactions (Figure 1). It integrates both structures and corresponding sequences of known protein-DNA complexes to learn an interpretable energy model based on the interacting amino acids and nucleotides at the protein-DNA binding interface. The model is trained using available protein-DNA complexes curated from existing databases [17, 18].

Unlike existing deep-learning-based protein-DNA binding prediction models, IDEA aims to learn a physicochemical-based energy model that quantitatively characterizes sequence-specific interactions between amino acids and nucleotides, thereby interpreting the “molecular grammar” driving the binding energetics of protein-DNA interactions. The optimized energy model can be used to predict the binding affinity of any given protein-DNA pair based on its structures and sequences. Additionally, it enables the prediction of genomic DNA binding sites by a given protein, such as a transcription factor. Finally, the learned energy model can be incorporated into a simulation framework to study the dynamics of DNA-binding processes, revealing mechanistic insights into various DNA-templated processes. Further details of the optimization protocol are provided in Methods Section Energy Model Optimization.”

The revised text in Section: Discussion now reads:

“Another highlight of IDEA is its ability to present an interpretable, familyspecific amino acid-nucleotide interaction energy model for given proteinDNA complexes. The optimized IDEA energy model can not only predict sequence-specific binding affinities of protein-DNA pairs but also provide a residue-specific interaction matrix that dictates the preferences of amino acidnucleotide interactions within specific protein families (Figure S11). This interpretable energy matrix would facilitate the discovery of sequence binding motifs for target DNA-binding proteins, complementing both sequencebased [24, 16, 25] and structure-based approaches [10, 26, 4, 15]. Additionally, we integrated this physicochemical-based energy model into a simulation framework, thereby improving the characterization of protein-DNA binding dynamics. IDEA-based simulation enables the investigation into dynamic interactions between various proteins and DNA, facilitating molecular-level understanding of the physical mechanisms underlying many DNA-binding processes, such as transcription, epigenetic regulations, and their modulation by sequence variations, such as single-nucleotide polymorphisms (SNPs) [22, 23].”

Comment 5: The manuscript disregards subtle yet important differences in commonly used terminology in the field. For example, the authors use the term ”specificity” and ”affinity” almost interchangeably (for example, the caption for Figure 3A uses ”specificity” although the Methods text describes the prediction as about ”affinity”). If the authors are looking to predict specificity, IDEA needs to be put in the context of the corresponding state-of-the-art (PMID 36123148, 39103447, 38867914, 36124796, etc).

We really appreciate the reviewer for pointing out the conflation of “specificity” and “affinity” in our manuscript. To clarify, the primary function of IDEA is to predict the binding affinities of protein-DNA pairs in a sequence-specific manner. We have revised the text to clarify the distinction between affinity and specificity and acknowledge prior works, including those provided by the reviewers, that focus on predicting protein-DNA binding specificity.

We have revised the Section title IDEA Accurately Predicts Protein-DNA Binding Specificity to IDEA Accurately Predicts Sequence-Specific Protein-DNA Binding Affinity; and ResidueLevel Protein-DNA Energy Model for Predicting Protein-DNA Recognition Specificities to Predictive Protein-DNA Energy Model at Residue Resolution.

We have revised the text in Introduction. The revised text reads:

“Computational methods complement experimental efforts by providing the initial filter for assessing sequence-specific protein-DNA binding affinity. Numerous methods have emerged to enable predictions of binding sites and affinities of DNA-binding proteins [27, 9, 1, 5, 28, 29, 30, 31, 8]. These methods often utilized machine-learning-based training to extract sequence preference information from DNA or protein by utilizing experimental high-throughput (HT) assays [27, 9, 1, 5, 28, 8], which rely on the availability and quality of experimental binding assays. Additionally, many approaches employ deep neural networks [29, 30, 31], which could obscure the interpretation of interaction patterns governing protein-DNA binding specificities. Understanding these patterns, however, is crucial for elucidating the molecular mechanisms underlying various DNA-recognition processes, such as those seen in TFs [32].”

We have revised the text in Section: IDEA Demonstrates Transferability across Proteins in the Same CATH Superfamily.

The revised text reads:

“Since IDEA relies on the sequence-structure relationship of given protein-DNA complexes to reach predictive accuracy, we inquired whether the trained energy model from one protein-DNA complex could be generalized to predict the sequence-specific binding affinities of other complexes. To test this, we assessed the transferability of IDEA predictions across all 11 structurally available protein-DNA complexes within the MAX TF-associated CATH superfamily (CATH ID: 4.10.280.10, Helix-loop-helix DNA-binding domain). We trained IDEA based on each of these 11 complexes and then used the trained model to predict the MAX-based MITOMI binding affinity. Our results show that IDEA generally makes correct predictions of the binding affinity when trained on proteins that are homologous to MAX, with Pearson and Spearman Correlation coefficients larger than 0.5 (Figure 3A and Figure S10).”

We have revised the caption of Figure 3: The revised text reads:

“IDEA prediction shows transferability within the same CATH superfamily. (A) The predicted MAX binding affinity, trained on other protein-DNA complexes within the same protein CATH superfamily, correlates well with experimental measurement. The proteins are ordered by their probability of being homologous to the MAX protein, determined using HHpred [33]. Training with a homologous protein (determined as a hit by HHpred) usually leads to better predictive performance (Pearson Correlation coefficient > 0.5) compared to non-homologous proteins. (B) Structural alignment between 1HLO (white) and 1A0A (blue), two protein-DNA complexes within the same CATH Helix-loop-helix superfamily. The alignment was performed based on the Ebox region of the DNA [34]. (C) The optimized energy model for 1A0A, a protein-DNA complex structure of the transcription factor PHO4 and DNA, with 33.41% probability of being homologous to the MAX protein. The optimized energy model is presented in reduced units, as explained in the Methods Section: Training Protocol.”

We have revised the text in Section Discussion: The revised text now reads:

“The protein-DNA interaction landscape has evolved to facilitate precise targeting of proteins towards their functional binding sites, which underlie essential processes in controlling gene expression. These interaction specifics are determined by physicochemical interactions between amino acids and nucleotides. By integrating sequences and structural data from available proteinDNA complexes into an interaction matrix, we introduce IDEA, a data-driven method that optimizes a system-specific energy model. This model enables high-throughput in silico predictions of protein-DNA binding specificities and can be scaled up to predict genomic binding sites of DNA-binding proteins, such as TFs. IDEA achieves accurate de novo predictions using only proteinDNA complex structures and their associated sequences, but its accuracy can be further enhanced by incorporating available experimental data from other binding assay measurements, such as the SELEX data [35, 36, 37], achieving accuracy comparable or better than state-of-the-art methods (Figures S2 and S7, Table S1 and S2). Despite significant progress in genome-wide sequencing techniques [38, 39, 40, 41], determining sequence-specific binding affinities of DNA-binding biomolecules remains time-consuming and expensive. Therefore, IDEA presents a cost-effective alternative for generating the initial predictions before pursuing further experimental refinement.”

We have revised the text in Discussion to clarify that the acquired binding affinities of target DNA sequences can be used to help existing models to infer specific DNA binding motifs.

The revised text now reads:

Another highlight of IDEA is its ability to present an interpretable, familyspecific amino acid-nucleotide interaction energy model for given proteinDNA complexes. The optimized IDEA energy model can not only predict sequence-specific binding affinities of protein-DNA pairs but also provide a residue-specific interaction matrix that dictates the preferences of amino acidnucleotide interactions within specific protein families (Figure S11). This interpretable energy matrix would facilitate the discovery of sequence binding motifs for target DNA-binding proteins, complementing both sequencebased [24, 16, 25] and structure-based approaches [10, 26, 4, 15]. Additionally, we integrated this physicochemical-based energy model into a simulation framework, thereby improving the characterization of protein-DNA binding dynamics. IDEA-based simulation enables the investigation into dynamic interactions between various proteins and DNA, facilitating molecular-level understanding of the physical mechanisms underlying many DNA-binding processes, such as transcription, epigenetic regulations, and their modulation by sequence variations, such as single-nucleotide polymorphisms (SNPs) [22, 23].

Comment 6: It is not clear how much the learned energy model is dependent on the structural model used for a specific system/family. It would be interesting to see the differences in learned model based on different representative PDB structures used. Similarly, the supplementary figures show a lack of discriminative power for proteins like PDX1 (homeodomain family), POU, etc. Can the authors shed some light on why such different performances?

We thank the reviewer for the insightful comments and agree that the trained energy model should be presented in the context of protein families. To further analyze the dependence of the energy model on protein family, we visualized the trained energy models for 24 proteins, including all proteins from the HT-SELEX dataset as well as PHO4 (PDB ID: 1A0A) and CTCF (PDB ID: 8SSQ), spanning 12 distinct protein families. To quantitatively assess similarities and differences among these energy models, we flattened each normalized energy model into an 80-dimensional vector and performed principal component analysis (PCA). As shown in Author response image 1 and Figure S11, energy models optimized from the same protein family fall within the same cluster, while those from different protein families exhibit distinct patterns. Moreover, the relative distance between energy models in PCA space reflects the degree of transferability. For example, PHO4 (PDB ID: 1A0A) is positioned close to MAX (PDB ID: 1HLO), whereas USF1 (PDB ID: 1AN4) and TCF4 (PDB ID: 6OD3) are farther away. This is consistent with the results shown in Figure 3A, where the energy model trained from PHO4 has better transferability than those from the other two systems.

We also greatly appreciate the reviewer’s suggestion to examine cases where IDEA failed to demonstrate strong discriminative power. When evaluating the model’s ability to distinguish between strong and weak binders, we used the available experimental structure most similar to the protein employed in the HT-SELEX experiments. In some instances, only the structure of the same protein from a different organism is available. For example, the HT-SELEX data for PDX1-DNA used the human PDX1 protein, but no human PDX1–DNA complex structure is available. Therefore, we used the mouse PDX1–DNA complex (PDB ID: 2H1K) for model training. The differences between species may limit the predictive accuracy of the model. A similar limitation applies to POU3F1, where an available mouse complex (PDB ID: 4Y60) was used to predict human protein–DNA interactions. Notably, DeepBind [9], a sequence-based prediction tool, also failed to distinguish strong from weak binders when using the mouse POU3F1 protein (AUC score: 0.457), but this was corrected with the human POU3F1 protein (AUC score: 0.956).

We also examined the remaining cases where IDEA did not show a clear distinction between strong and weak binders: USF1, Egr1, and PROX1. For PROX1, we initially used the structure of a protein-DNA complex (PDB ID: 4Y60) in training. However, upon closer inspection, we discovered that this structure does not include the PROX1 protein, but SOX-18, a different transcription factor. This explains the inaccurate prediction made by IDEA. Since no experimental PROX1-DNA complex structure is currently available, we have removed this case from our HT-SELEX evaluation.

IDEA also fails to fully resolve the binding preference of USF1. A closer examination of the HT-SELEX data reveals a lack of distinction among the sequences, as most sequences, including those with the lowest M-word (binding affinity) scores, contain the DNA-binding E-box sequence CACGTG. Therefore, USF1 represents a challenging example where the experimental data only consists of strong binders with limited variations in binding affinity, which likely results from differences in flanking sequences of the E-box motif.

Egr1 stands as a peculiar example. Whereas IDEA does not effectively distinguish between the strong and weak binders in the current HT-SELEX dataset, its predictions are consistent with other experimental datasets, including binding affinities measured by kMITOMI [42] (Figure S8A, B), preferred binding sequences from protein-binding microarray, an earlier HT-SELEX experiment, and bacterial one-hybrid data [43]. Therefore, further investigation of the current HT-SELEX data is needed to reconcile these differences.

We have included additional text in Section: IDEA Demonstrates Transferability across Proteins in the Same CATH Superfamily to discuss the PCA analysis and the dependence of the model’s transferability on the similarity among the learned energy models.

The revised text now reads:

“The transferability of IDEA within the same CATH superfamily can be understood from the similarities in protein-DNA binding interfaces, which determine similar learned energy models. For example, the PHO4 protein (PDB I”D: 1A0A) shares a highly similar DNA-binding interface with the MAX protein (PDB ID: 1HLO) (Figure 3B), despite sharing only a 33.41% probability of being homologous. Consequently, the energy model derived from the PHO4DNA complex (Figure 3C) exhibits a similar amino-acid-nucleotide interactive pattern as that learned from the MAX-DNA complex (Figure 2B). To further evaluate the similarity between the learned energy models and their connection to protein families, we performed principal component analysis (PCA) on the normalized energy models across 24 proteins from 12 protein families [5]. Our analysis (Figure S11) reveals that most of the energy models from the same protein family fall within the same cluster, while those from different protein families exhibit distinct patterns. Moreover, the relative distance between energy models in PCA space reflects the degree of transferability between them. For example, PHO4 (PDB ID: 1A0A) is positioned close to MAX (PDB ID: 1HLO), whereas USF1 (PDB ID: 1AN4) and TCF4 (PDB ID: 6OD3) are farther away. This is consistent with the results in Figure 3A, where the energy model trained on PHO4 has better transferability than those trained on USF1 or TCF4.”

We have also added an Appendix section titled Analysis of examples where IDEA fails to recognize strong DNA binders to discuss the examples in which IDEA did not perform well:

“We examine IDEA’s capability in identifying strong binders from the HT-SELEX dataset across 12 protein families [5]. The model successfully predicts 18 out of 22 protein-DNA systems, but the performance is reduced in 4 cases. Closer investigations revealed the source of these limitations. In some instances, only the protein from a different organism is available. For example, the PDX1 HT-SELEX data utilized the human PDX1 protein, but no human PDX1–DNA complex structure is available. Therefore, the mouse PDX1–DNA complex structure (PDB ID: 2H1K) was used for model training. Differences between model organisms may reduce predictive accuracy. A similar limitation applies to POU3F1, where an available mouse complex (PDB ID: 4Y60) was used to predict human protein–DNA interactions. Notably, DeepBind [9], a sequence-based prediction tool, also failed to distinguish strong from weak binders when using the mouse POU3F1 protein (AUC score: 0.457), but this was corrected with the human POU3F1 protein (AUC score: 0.956).

IDEA also fails to fully resolve the binding preference of USF1. A closer examination of the HT-SELEX data reveals a lack of distinction among the sequences, as most sequences, including those with the lowest M-word (binding affinity) scores, contain the DNA-binding E-box sequence CACGTG. Therefore, USF1 represents a challenging example where the experimental data only consists of strong binders with limited variations in binding affinity, which likely results from differences in flanking sequences of the E-box motif.

Egr1 stands as a peculiar example. Whereas IDEA does not effectively distinguish between the strong and weak binders in the current HT-SELEX dataset, its predictions are consistent with other experimental datasets, including binding affinities measured by k-MITOMI [42] (Figure S8A, B), preferred binding sequences from protein-binding microarray, an earlier HT-SELEX experiment, and bacterial one-hybrid data [43]. Therefore, further investigation of the current HT-SELEX data is needed to reconcile these differences.”

Comment 7: It is also not clear if IDEA’s prediction for reverse complement sequences is the same for a given sequence. If so, how is this property being modelled? Either this description is lacking or I missed it.

We thank the reviewer for the insightful comments. Given a target protein-DNA sequence, the IDEA protocol substitutes it into a known protein-DNA complex structure to evaluate the binding free energy, which can be converted into binding affinity. IDEA uses sequence identity to determine whether the forward or reverse strand of the DNA should be replaced. Only the strand most similar to the target sequence is substituted. As a result, the model treats reverse-complement sequences differently. As the orientations of test sequences are specified from 5’ to 3’ in all datasets used in this study (e.g., processed MITOMI, HT-SELEX, and ChIP-seq data), this approach ensures that the target sequences are replaced and evaluated correctly. In cases where sequence orientation is not provided (though this was not an issue in this study), we recommend replacing both the forward and reverse strands with the target sequence separately and evaluating the corresponding protein–DNA binding free energies. Since strong binders are likely to dominate the experimental signals, the higher predicted binding affinity, with stronger binding free energies, should be taken as the model’s final prediction.

We have added one section to the Methods Section titled Treatment of Complementary DNA Sequences to clarify these modeling details.

The specific text reads:

To replace the DNA sequence in the protein-DNA complex structure with a target sequence, IDEA uses sequence identity to determine whether the target sequence belongs to the forward or reverse strand of the DNA in the proteinDNA structure. The more similar strand is selected and replaced with the target sequence. As the orientations of test sequences are specified from 5’ to 3’ in all datasets used in this study (e.g., processed MITOMI, HT-SELEX, and ChIP-seq data), this approach ensures that the target sequences are replaced and evaluated correctly. In cases where sequence orientation is not provided (though this was not an issue in this study), we recommend replacing both the forward and reverse strands with the target sequence separately and evaluating the corresponding protein–DNA binding free energies. Since strong binders are likely to dominate the experimental signals, the higher predicted binding affinity, with stronger binding free energy, should be taken as the model’s final prediction.”

“Comment 8: Page 21 line 403, the E-box core should be CACGTG instead of CACGTC.

We apologize for our oversight and have corrected the relevant text.

Comment 9: The citation for DNAproDB is outdated and should be updated (PMID 39494533).

We thank the reviewer for pointing this out and have updated our citation accordingly.

**Reviewer #3:**
Comment 0: Summary: Protein-DNA interactions and sequence readout represent a challenging and rapidly evolving field of study. Recognizing the complexity of this task, the authors have developed a compact and elegant model. They have applied well-established approaches to address a difficult problem, effectively enhancing the information extracted from sparse contact maps by integrating artificial sequences decoy set and available experimental data. This has resulted in the creation of a practical tool that can be adapted for use with other proteins.

We appreciate the reviewer’s excellent summary of the paper, and we thank the reviewer for the insightful suggestions and comments.

Comment 1: Strengths: (1) The authors integrate sparse information with available experimental data to construct a model whose utility extends beyond the limited set of structures used for training. (2) A comprehensive methods section is included, ensuring that the work can be reproduced. Additionally, the authors have shared their model as a GitHub project, reflecting their commitment to transparency of research.

We appreciate the reviewer’s strong assessment of the strengths of this paper. In addition to sharing our model on GitHub, we have also uploaded the original data and the essential scripts required to reproduce the results presented in the manuscript. We hope this further demonstrates our commitment to transparency and reproducibility.

Comment 2: Weaknesses: (1) The coarse-graining procedure appears artificial, if not confusing, given that full-atom crystal structures provide more detailed information about residue-residue contacts. While the selection procedure for distance threshold values is explained, the overall motivation for adopting this approach remains unclear. Furthermore, since this model is later employed as an empirical potential for molecular modeling, the use of P and C5 atoms raises concerns, as the interactions in 3SPN are modeled between Cα and the nucleic base, represented by its center of mass rather than P or C5 atoms.

We appreciate the reviewer’s insightful comments. The selection of P and C5 atoms was based on different relative positions of protein and DNA across various complex structures, each with distinctive protein-DNA structural interfaces. To illustrate this, we selected two representative structures where our algorithm selected C5 and P atoms, respectively: MAX-DNA (PDB ID: 1HLO) and FOXP3 (PDB ID: 7TDW). As shown in Author response image 2, in the case of 1HLO, more C5 atoms are within the cutoff distance of 10 A from° the protein Cα atoms, thus capturing essential contacting interactions. In contrast, 7TDW has more P atoms within this cutoff. Importantly, several P atoms are distributed on the minor groove of the DNA, which were not captured by the C5 atoms. To maximize the inclusion of relevant structural contacts, we employed a filtering scheme that selectively chooses either P or C5 atoms based on their proximity to the protein to enhance the model prediction. We note that while this scheme is helpful, the IDEA predictions remain robust across different atom selections. To assess this robustness, we performed binding affinity predictions using only P atoms on the HT-SELEX dataset across 12 protein families [5]. Our predictions (Author response table 1) show comparable performance to that achieved using our filtering scheme.

**Author response image 2. sa3fig2:** Comparison between P and C5 atoms in proximity to the protein 3D structures of MAX–DNA (A) and FOXP-DNA (B) complexes, where P atoms (red sphere) and C5 atoms (blue sphere) that are within 10 A of Cα atoms are highlighted.

When incorporating the trained IDEA energy model into a simulation model, we acknowledge a potential mismatch between the resolution of the data-driven model (one coarse-grained site per nucleotide) and the 3SPN simulation model (three coarse-grained sites per nucleotide). The selection of nucleic base sites for molecular interactions in the 3SPN model follows our previous work [44] and its associated code implementation. While revisiting this part of the manuscript, we identified an inconsistency in the reported results in Figure 5A of our initial version: Specifically, we previously used the protein side-chain atoms, rather than only the Cα atoms, in model training. Retraining the data using the Cα atoms results in reduced prediction performance for the IDEA model (Figure 5A). Nonetheless, incorporating this updated energy model into simulations still yielded high accuracy in the predicted absolute binding free energies (Author response image 3A), demonstrating the robustness of our simulation framework in predicting absolute binding free energies against variations in atom selection during the IDEA model training. Following the reviewer’s suggestion, we also incorporated the IDEA-trained energy model as short-range van der Waals interactions between protein Cα atoms and DNA P atoms. As shown in Author response image 3B, our simulation reveals a slightly improved performance over our original implementation, with higher Pearson and Spearman correlation coefficients and a fitted slope closer to 1.0. This result suggests that a more consistent atom selection scheme between the data-driven and simulation models can improve the overall predictions. Accordingly, we have updated Figure 5 with this improved setup, using the simulation model with short-range vdW interactions implemented between protein Cα atoms and DNA P atoms (Figure 5C), ensuring consistency between the IDEA model and simulation framework.

**Author response table 1. sa3table1:** Comparison of IDEA performance using two DNA atom selection schemes: the filtering scheme presented in the manuscript (C5 and P atoms) versus using only P atoms. Cases where the two schemes result in different atom selections are highlighted in bold.

Protein	AUC (C5 & P)	AUC (P only)	PRAUC (C5 & P)	PRAUC (P only)
ELK1	0.810	1.000	0.651	1.000
FOXP3	0.873	0.873	0.738	0.738
MAX	0.941	0.879	0.772	0.897
TCF4	0.638	0.638	0.617	0.617
USF1	0.484	0.484	0.474	0.474
Cebpb	0.983	0.989	0.976	0.942
CEBPB	0.923	0.931	0.691	0.699
Mafb	0.607	0.607	0.539	0.539
Egr1	0.181	0.181	0.275	0.275
YY1	1.000	1.000	1.000	1.000
ZBTB7A	0.998	0.998	0.830	0.830
GATA3	0.816	0.816	0.636	0.636
ALX4	0.631	0.631	0.542	0.542
BARHL2	0.822	0.822	0.737	0.737
MSX1	0.926	0.926	0.815	0.815
PDX1	0.595	0.595	0.488	0.488
HSF1	0.993	0.993	0.992	0.992
SMAD3	0.992	0.992	0.903	0.903
MEIS1	0.797	0.797	0.631	0.631
NR2C2	1.000	1.000	1.000	1.000
NR3C1	0.921	0.921	0.791	0.791
POU3F1	0.324	0.220	0.338	0.338

We acknowledge that a gap still exists between the resolution of the data-driven and simulation models. To ensure a completely consistent coarse-grained level between these two models, we will work on implementing the IDEA model output for 1-bead-per-nucleotide DNA simulation models in the future.

Comment 3: (2) Although the authors use a standard set of metrics to assess model quality and predictive power, some ∆∆G predictions compared to MITOMI-derived ∆∆G values appear nonlinear, which casts doubt on the interpretation of the correlation coefficient.

**Author response image 3. sa3fig3:** Comparison of simulations using different representative atoms. (A) Protein-DNA binding simulation with the IDEA-model incorporated as short-range van der Waals between protein Cα atom and nucleic base site. (B) Protein-DNA binding simulation with the IDEA-model incorporated as short-range van der Waals between protein Cα atom and DNA P atoms. The predicted free energies are robust to the choice of DNA representative atoms. The predicted binding free energies are presented in physical units, and error bars represent the standard deviation of the mean.

We thank the reviewer for the insightful comments and agree that the linear fit between our model’s prediction and the experimental data may not be the best measure of performance. The primary utility of the IDEA model is to predict high-affinity DNA-binding sequences for a given DNA-binding protein by assessing the relative binding affinities across different DNA sequences. In this regard, the ranked order of predicted sequence binding affinities serves as a better metric for evaluating the success of this model. To evaluate this, we calculated both Spearman’s rank correlation coefficient, which does not rely on linear correlation, and the Pearson correlation coefficient between our predictions and the experimental results. As shown in Figure 2, our computation shows a Spearman’s rank correlation coefficient of 0.65 for the MAX-based predictions using one MAX-DNA complex (PDB ID: 1HLO), supporting the model’s capability to effectively distinguish strong from weak binders.

As reflected in Figure 2 of the main text, although our model generally captures the relative binding affinities across different DNA sequences, its predictive accuracy diminishes for low-affinity sequences (Figure 2). This could be due to two limitations of the current modeling framework: (1) The model is residue-based and estimates binding free energy as the additive sum of contributions from individual contacting amino-acid-nucleotide pairs. This assumption does not account for cooperative effects caused by simultaneous changes at multiple nucleotide positions. One potential direction to further improve the model would be to use a finer-grained representation by incorporating more atom types within contacting residues, and to use a many-body potential to better capture cooperative effects from multiple mutations. (2) The model assumes that the target DNA adopts the same binding interface as in the reference crystal structure. However, sequencedependent DNA shape has been shown to be important in determining protein-DNA binding affinity [1]. To address this limitation, a future direction is to use deep-learningbased methods to incorporate predicted DNA shape or protein-DNA complex structures based on their sequences [2, 3] into our model prediction.

To fully evaluate the predictive power of IDEA, we have included Spearman’s rank correlation coefficient for every correlation plot in this manuscript. Across all our analyses, the Spearman’s rank correlation coefficients reveal similar predictive performance as the Pearson correlation coefficients. Additionally, we have included in our discussion the current limitations of our model and potential directions for future improvement.

We have edited our Discussion Section to include a discussion on the limitations of the current model. Specifically, the added texts are:

“Although IDEA has proved successful in many examples, it can be improved in several aspects. The model currently assumes the training and testing sequences share the same protein-DNA structure. While double-stranded DNA is generally rigid, recent studies have shown that sequence-dependent DNA shape contributes to their binding specificity [1, 2, 4]. To improve predictive accuracy, one could incorporate predicted DNA shapes or structures into the IDEA training protocol. In addition, the model is residue-based and evaluates the binding free energy as the additive sum of contributions from individual amino-acid-nucleotide contacts. This assumption does not account for cooperative effects that may arise from multiple nucleotide changes. A potential refinement could utilize a finer-grained model that includes more atom types within contacting residues and employs a many-body potential to account for such cooperative effects.”

Comment 4: (3) The discussion section lacks information about the model’s limitations and a comprehensive comparison with other models. Additionally, differences in model performance across various proteins and their respective predictive powers are not addressed.

We thank the reviewer for the insightful comments. As discussed in the response to Comment 3, the current structural model has several limitations, which may reduce predictive accuracy for weak DNA binders. We have noted these limitations in the Discussion section.

To compare the performance of IDEA with state-of-the-art protein-DNA predictive models, we examined the predictive accuracies of two additional popular computational models: ProBound [8] and DeepBind [9]. ProBound has been shown to have a better performance than several earlier predictive protein-DNA models, including JASPAR 2018 [11], HOCOMOCO [12], Jolma et al. [13], and DeepSELEX [14]. To benchmark these models’ performance, we examine each method’s capability to identify strong binders with the HT-SELEX datasets covering 22 proteins from 12 protein families [5]. As suggested by Reviewer 1, we also calculated the PRAUC score, reweighted to account for data imbalance [6], as a complementary metric for evaluating the model performance.

As shown in Figure S6, Table S1, and Table S2, IDEA ranked second among the three predictive methods. It is important to note that both ProBound and DeepBind were trained on a curated version of the HT-SELEX data [13], which overlaps with the testing data [5]. Compared with them, IDEA was trained only on the given structural and sequence information from a single protein-DNA complex, thus independent of the testing data. In order to assess how IDEA performs when incorporating knowledge from HT-SELEX data, we augmented the training by randomly including half of the HT-SELEX data (see the Methods Section Enhanced Modeling Prediction with SELEX Data). The augmented IDEA model achieved the best performance among all the models. We further benchmarked IDEA using a 10-fold cross-validation on the same HT-SELEX data [5] and found that IDEA outperformed a recent regression model that considers the shape of DNA with different sequences [5]. Overall, IDEA can be used to predict protein-DNA affinities in the absence of known binding sequence data, thereby filling a critical gap when such experimental datasets are unavailable.

In addition, we compared the performance of IDEA with both general and family-specific knowledge-based energy models. First, we incorporated a knowledge-based generic protein-DNA energy model (DBD-Hunter) learned from the protein-DNA database, reported by Skoinick and coworkers [7], into our prediction protocol. This model assigns interaction energies to different functional groups within each DNA nucleotide (e.g., phosphate (PP), sugar (SU), pyrimidine (PY), and imidazole (IM) groups). For our comparison, we averaged the energy contributions of these groups within each nucleotide and replaced the IDEA-learned energy model with this generic one to test its ability to differentiate strong binders from weak binders in the HT-SELEX dataset [5]. As shown in Figure S6, the IDEA model generally achieves better performance than the generic energy model. Additionally, we compared IDEA with rCLAMPS, a family-specific energy model developed to predict protein-DNA binding specificity in the C2H2 and homeodomain families. As shown in Table S1 and Table S2, IDEA also shows better performance than rCLAMPS in most cases across the C2H2 and homeodomain families, demonstrating that it has better predictive accuracy than both family-specific and generic knowledge-based models.

We have revised our text to include the comparison between IDEA and other predictive models. Specifically, we revised the text in Section: IDEA Generalizes across Various Protein Families.

The revised text reads:

“To examine IDEA’s predictive accuracy across different DNA-binding protein families, we applied it to calculate protein-DNA binding affinities using a comprehensive HT-SELEX dataset [5]. We focused on evaluating the capability of IDEA to distinguish strong binders from weak binders for each protein with an experimentally determined structure. We calculated the probability density distribution of the top and bottom binders identified in the SELEX experiment. A well-separated distribution indicates the successful identification of strong binders by IDEA (Figure 2D and S4). Receiver Operating Characteristic (ROC) analysis was performed to calculate the Area Under the Curve (AUC) and the precision-recall curve (PRAUC) scores for these predictions. Further details are provided in the Methods Section Evaluation of IDEA Prediction Using HT-SELEX Data. Our analysis shows that IDEA successfully differentiates strong from weak binders for 80% of the 22 proteins across 12 protein families, achieving AUC and balanced PRAUC scores greater than 0.5 (Figure 2E and S5). To benchmark IDEA’s performance against other leading methods, we compared its predictions with several popular models, including the sequence-based predictive models ProBound [8] and DeepBind [9], the familybased energy model rCLAMPS [10], and the knowledge-based energy model DBD-Hunter [7]. IDEA demonstrates performance comparable to these stateof-the-art approaches (Figure S6, Table S1, and Table S2), and incorporating sequence features further improves its prediction accuracy. We also performed 10-fold cross-validation on the binding affinities of protein–DNA pairs in this dataset and found that IDEA outperforms a recent regression model that considers the shape of DNA with different sequences [5] (Figure S7). Details are provided in Section: Comparison of IDEA predictive performance Using HT-SELEX data.”

We also added one section Comparison of IDEA predictive performance Using HT-SELEX data in the Appendix to fully explain the comparison between IDEA and other popular models.

The added texts are:

“To benchmark the performance of IDEA against state-of-the-art protein-DNA predictive models, we evaluated its ability to recognize strong binders with the HT-SELEX datasets across 22 proteins from 12 families [5]. Specifically, we compare IDEA with two widely used sequence-based models: ProBound [8] and DeepBind [9]. ProBound has demonstrated superior performance over many other predictive protein-DNA models, including JASPAR 2018 [11], HOCOMOCO [12], Jolma et al. [13], and DeepSELEX [14]. To use ProBound, we retrieved the trained binding model for each protein from motifcentral.org and used the GitHub implementation of ProBoundTools to infer the binding scores between protein and target DNA sequences. Except for POU3F1, binding models are available for all proteins. Therefore, we excluded POU3F1 and evaluated the protein-DNA binding affinities for the remaining 21 proteins. To use DeepBind, sequence-specific binding affinities were predicted directly with its web server. The Area Under the Curve (AUC) and the Precision-Recall AUC (PRAUC) scores were used as metrics for comparison. An AUC score of 1.0 indicates a perfect separation between the strong- and weak-binder distributions, while an AUC score of 0.5 indicates no separation. Because there is a significant imbalance in the number of strong and weak binders from the experimental data [5], where the strong binders are far fewer than the weak binders, we reweighted the samples to achieve a balanced evaluation, using 0.5 as the baseline for randomized prediction [6]. As summarized in Figure S6, Table S1, and Table S2, IDEA ranked second among the three predictive models. In order to assess the performance of IDEA when augmented with additional protein-DNA binding data, we augmented IDEA using randomly selected half of the HT-SELEX data (see the Methods Section Enhanced Modeling Prediction with SELEX Data). The augmented IDEA model achieved the best performance among all the models.”

“In addition, we compared the performance of IDEA with both general and family-specific knowledge-based energy models. First, we incorporated a knowledgebased generic protein-DNA energy model (DBD-Hunter) learned from the protein-DNA database, reported by Skoinick and coworkers [7], into our prediction protocol. This model assigns interaction energies to different functional groups within each DNA nucleotide, including phosphate (PP), sugar (SU), pyrimidine (PY), and imidazole (IM) groups. For our comparison, we averaged the energy contributions of these groups within each nucleotide and replaced the IDEA-learned energy model with the DBD-Hunter model to assess its ability to differentiate strong binders from weak binders in the HTSELEX dataset [5]. Additionally, we compared IDEA with rCLAMPS, a familyspecific energy model developed to predict protein-DNA binding specificity in the C2H2 and homeodomain families. rCLAMPS learns a position-dependent amino-acid-nucleotide interaction energy model. To incorporate this model into the binding free energy calculation, we averaged the energy contributions across all occurrences of each amino-acid-nucleotide pair, which resulted in a 20-by-4 residue-type-specific energy matrix. This matrix is structurally analogous to the IDEA-trained energy model and can be directly integrated into the binding free energy calculations. As shown in Figure S6, Table S1, and Table S2, the IDEA model generally outperforms DBD-Hunter and rCLAMPS, demonstrating that it can achieve better predictive accuracy than both generic and family-specific knowledge-based models.”

“We also performed 10-fold cross-validation using the same HT-SELEX datasets, following the protocol described in the Methods Section Enhanced Modeling Prediction with SELEX Data. For each protein, we divided the entire dataset into 10 equal, randomly assigned folds. In each iteration, we used randomly selected 9 of the 10 folds as the training dataset and the remaining fold as the testing dataset. This process was repeated 10 times so that each fold served as the test set once. We then reported the average R2 scores across these iterations to evaluate IDEA’s predictive performance. Our results are compared with the 1mer and 1mer+shape methods from [5], the latest regression model that considers the shape of DNA with different sequences (Figure S7). This comparative analysis shows IDEA achieved higher predictive accuracy than the state-of-the-art sequence-based protein-DNA binding predictors for proteinDNA complexes that have available experimentally resolved structures.”

“Overall, these results demonstrate that IDEA can be used to predict the proteinDNA pairs in the absence of known binding sequence data, thus filling an important gap in protein-DNA predictions when experimental binding sequence data are unavailable.”

We also greatly appreciate the reviewer’s suggestion to examine the model’s performance across different proteins. To do this, we first evaluated the dependence of IDEA prediction on the availability of experimental structures similar to the target protein-DNA complexes. To quantitatively assess similarities and differences among the IDEA-derived energy models, we flattened each normalized energy model into an 80-dimensional vector and performed principal component analysis (PCA). As shown in Author response image 1 and Figure S11, energy models optimized from the same protein family fall within the same cluster, while those from different protein families exhibit distinct patterns. Moreover, the relative distance between energy models in PCA space reflects the degree of transferability. For example, PHO4 (PDB ID: 1A0A) is positioned close to MAX (PDB ID: 1HLO), whereas USF1 (PDB ID: 1AN4) and TCF4 (PDB ID: 6OD3) are farther away. This is consistent with the results shown in Figure 3A, where the energy model trained from PHO4 has better transferability than those from the other two systems. Therefore, the availability of experimental structures from protein-DNA complexes more similar to the target can lead to better predictive performance.

We also examine cases in which the IDEA model failed to show strong discriminative power for protein-DNA complexes in the HT-SELEX datasets [5] (Figures 2E and S5). When evaluating the model’s ability to distinguish between strong and weak binders, we used the available experimental structure most similar to the protein employed in the HT-SELEX experiments. In some instances, only the structure of the same protein from a different organism is available. For example, the HT-SELEX data for PDX1-DNA used the human PDX1 protein, but no human PDX1–DNA complex structure is available. Therefore, we used the mouse PDX1–DNA complex (PDB ID: 2H1K) for model training. The differences between species may limit the predictive accuracy of the model. A similar limitation applies to POU3F1, where an available mouse complex (PDB ID: 4Y60) was used to predict human protein–DNA interactions. Notably, DeepBind [9], a sequencebased prediction tool, also failed to distinguish strong from weak binders when using the mouse POU3F1 protein (AUC score: 0.457), but this was corrected with the human POU3F1 protein (AUC score: 0.956).

We also examined the remaining cases where IDEA did not show a clear distinction between strong and weak binders: USF1, Egr1, and PROX1. For PROX1, we initially used the structure of a protein-DNA complex (PDB ID: 4Y60) in training. However, upon closer inspection, we discovered that this structure does not include the PROX1 protein, but SOX-18, a different transcription factor. This explains the inaccurate prediction made by IDEA. Since no experimental PROX1-DNA complex structure is currently available, we have removed this case from our HT-SELEX evaluation.

IDEA also fails to fully resolve the binding preference of USF1. A closer examination of the HT-SELEX data reveals a lack of distinction among the sequences, as most sequences, including those with the lowest M-word (binding affinity) scores, contain the DNA-binding E-box sequence CACGTG. Therefore, USF1 represents a challenging example where the experimental data only consists of strong binders with limited variations in binding affinity, which likely results from differences in flanking sequences of the E-box motif.

Egr1 stands as a peculiar example. Whereas IDEA does not effectively distinguish between the strong and weak binders in the current HT-SELEX dataset, its predictions are consistent with other experimental datasets, including binding affinities measured by kMITOMI [42] (Figure S8A, B), preferred binding sequences from protein-binding microarray, an earlier HT-SELEX experiment, and bacterial one-hybrid data [43]. Therefore, further investigation of the current HT-SELEX data is needed to reconcile these differences.

In summary, IDEA’s predictive performance depends on the availability of experimental structures closely related to the target protein-DNA complexes, both in terms of protein sequences and model organisms.

We have included additional text in Section: IDEA Demonstrates Transferability across Proteins in the Same CATH Superfamily to discuss the PCA analysis and the dependence of the model’s transferability on the similarity among the learned energy models.

The revised text now reads:

“The transferability of IDEA within the same CATH superfamily can be understood from the similarities in protein-DNA binding interfaces, which determine similar learned energy models. For example, the PHO4 protein (PDB ID: 1A0A) shares a highly similar DNA-binding interface with the MAX protein (PDB ID: 1HLO) (Figure 3B), despite sharing only a 33.41% probability of being homologous. Consequently, the energy model derived from the PHO4DNA complex (Figure 3C) exhibits a similar amino-acid-nucleotide interactive pattern as that learned from the MAX-DNA complex (Figure 2B). To further evaluate the similarity between the learned energy models and their connection to protein families, we performed principal component analysis (PCA) on the normalized energy models across 24 proteins from 12 protein families [5]. Our analysis (Figure S11) reveals that most of the energy models from the same protein family fall within the same cluster, while those from different protein families exhibit distinct patterns. Moreover, the relative distance between energy models in PCA space reflects the degree of transferability between them. For example, PHO4 (PDB ID: 1A0A) is positioned close to MAX (PDB ID: 1HLO), whereas USF1 (PDB ID: 1AN4) and TCF4 (PDB ID: 6OD3) are farther away. This is consistent with the results in Figure 3A, where the energy model trained on PHO4 has better transferability than those trained on USF1 or TCF4.”

We have also added an Appendix section titled Analysis of examples where IDEA fails to recognize strong DNA binders to discuss the examples in which IDEA did not perform well:

“We examine IDEA’s capability in identifying strong binders from the HT-SELEX dataset across 12 protein families [5]. The model successfully predicts 18 out of 22 protein-DNA systems, but the performance is reduced in 4 cases. Closer investigations revealed the source of these limitations. In some instances, only the protein from a different organism is available. For example, the PDX1 HT-SELEX data utilized the human PDX1 protein, but no human PDX1–DNA complex structure is available. Therefore, the mouse PDX1–DNA complex structure (PDB ID: 2H1K) was used for model training. Differences between model organisms may reduce predictive accuracy. A similar limitation applies to POU3F1, where an available mouse complex (PDB ID: 4Y60) was used to predict human protein–DNA interactions. Notably, DeepBind [9], a sequence-based prediction tool, also failed to distinguish strong from weak binders when using the mouse POU3F1 protein (AUC score: 0.457), but this was corrected with the human POU3F1 protein (AUC score: 0.956).

IDEA also fails to fully resolve the binding preference of USF1. A closer examination of the HT-SELEX data reveals a lack of distinction among the sequences, as most sequences, including those with the lowest M-word (binding affinity) scores, contain the DNA-binding E-box sequence CACGTG. Therefore, USF1 represents a challenging example where the experimental data only consists of strong binders with limited variations in binding affinity, which likely results from differences in flanking sequences of the E-box motif.

Egr1 stands as a peculiar example. Whereas IDEA does not effectively distinguish between the strong and weak binders in the current HT-SELEX dataset, its predictions are consistent with other experimental datasets, including binding affinities measured by k-MITOMI [42] (Figure S8A, B), preferred binding sequences from protein-binding microarray, an earlier HT-SELEX experiment, and bacterial one-hybrid data [43]. Therefore, further investigation of the current HT-SELEX data is needed to reconcile these differences.”

Comment 5: The authors provide an implementation of their model via GitHub, which is commendable. However, it unexpectedly requires the Modeller suite, despite no details about homology modeling being included in the methods section.

We thank the reviewer for the helpful comments. We did not use the homology modeling module of Modeller. Instead, we only used a single Python script, buildseq.py, from the Modeller package to extract the protein and DNA sequences from the given PDB structure. We have clarified this in the README file on our GitHub repository.

Comment 6: While the manuscript is written in clear and accessible English, some sentences are quite long and could benefit from rephrasing (e.g., lines 49-52).

Thank you for the helpful suggestion. We agree that the original sentence was overly long and have revised it by splitting it into two for improved clarity and readability.

The revised version reads:

“The very robustness of evolution [46, 47, 48, 49] provides an opportunity to extract the sequence-structure relationships embedded in existing complexes. Guided by this principle, we can learn an interpretable binding energy landscape that governs the recognition processes of DNA-binding proteins.”

Comment 7: In line 82, the citations appear out of place, as the context seems to suggest the use of the newly developed model.

Thank you for this insightful suggestion. We have rephrased the sentence to better connect with the context of this section.

The revised text now reads:

“Finally, the learned energy model can be incorporated into a simulation framework to explore the dynamics of DNA-binding processes, revealing mechanistic insights into various DNA-templated processes.”

Comment 8: Line 143 ”different structure from the bHLH TFs and thus requires a different atom” This is the first instance in the manuscript where the atom selection for distance thresholding is mentioned, making the text somewhat confusing.

We thank the reviewer for the insightful comment and agree that the atom selection scheme appears abruptly in this section. To improve clarity, we have moved the detailed atom selection scheme and its rationale to the Methods Section titled Structural Modeling of Protein and DNA.

Comment 9: Figures: Overall, the figures are visually appealing but could be further improved.

We appreciate the positive feedback regarding the visual presentation of our figures. Following the reviewer’s suggestions and to further enhance clarity, we have revised several figures to improve labeling, layout, and annotations.

Comment 10: Figure 1: The description ”highlighted in blue” considers changing to ”highlighted in blue on the structure.”.

We have revised the text based on your suggestion.

Comment 11: Figure 2: Panel B is missing a color bar legend and units, as is the case in Figure 3C. Additionally, the placement of Panel C is unconventional - it appears it should be Panel D. The color scheme for the spheres is not fully described. Panel E: There are too many colors used; consider employing different markers to improve clarity.

Thank you for the helpful suggestions.

For Figure 2B and Figure 3C, we would like to clarify that the predicted energies are presented in reduced units due to an undetermined prefactor introduced during the model optimization. This point has now been clarified in the figure captions and is also explained in the Methods section titled Training Protocol.

Additionally, we have rearranged Panels C and D to improve the figure layout and have fully described the color coding used in the structural representations.

We have updated it to read:

“Results for MAX-based predictions. (A) The binding free energies calculated by IDEA, trained using a single MAX–DNA complex (PDB ID: 1HLO), correlate well with experimentally measured MAX–DNA binding free energies [50]. ∆∆G represents the changes in binding free energy relative to that of the wild-type protein–DNA complex. (B) The heatmap, derived from the optimized energy model, illustrates key amino acid–nucleotide interactions governing MAX–DNA recognition, showing pairwise interaction energies between 20 amino acids and the four DNA bases—DA (deoxyadenosine), DT (deoxythymidine), DC (deoxycytidine), and DG (deoxyguanosine). Both the predicted binding free energies and the optimized energy model are expressed in reduced units, as explained in the Methods Section Training Protocol. Each cell represents the optimized energy contribution, where blue indicates more favorable (lower) energy values, and red indicates less favorable (higher) values. (C) The 3D structure of the MAX–DNA complex (zoomed in with different views) highlights key amino acid–nucleotide contacts at the protein–DNA interface. Notably, several DNA deoxycytidines (red spheres) form close contacts with arginines (blue spheres). Additional nucleotide color coding: adenine (yellow spheres), guanine (green spheres), thymine (pink spheres). (D) Probability density distributions of predicted binding free energies for strong (blue) and weak (red) binders of the protein ZBTB7A. The mean of each distribution is marked with a dashed line. (E) Summary of AUC scores for protein–DNA pairs across 12 protein families, calculated based on the predicted probability distributions of binding free energies.”

We fully agree that Panel E was visually overwhelming. We have revised the plot by using a combination of color and marker shapes to more clearly distinguish between different protein families, as suggested.

Comment 12: Typos:Line 18: Gene expressions → Gene expression?Line 28: performed → utilized ?

We really appreciate the suggestions and have corrected the text accordingly.

References

(1) Tianyin Zhou, Ning Shen, Lin Yang, Namiko Abe, John Horton, Richard S Mann, Harmen J Bussemaker, Raluca Gordan, and Remo Rohs. Quantitative modeling ofˆ transcription factor binding specificities using DNA shape. Proceedings of the National Academy of Sciences, 112(15):4654–4659, 2015.

(2) Jinsen Li, Tsu-Pei Chiu, and Remo Rohs. Predicting DNA structure using a deep learning method. Nat Commun, 15(1):1243, February 2024.

(3) Josh Abramson, Jonas Adler, Jack Dunger, Richard Evans, Tim Green, Alexander Pritzel, Olaf Ronneberger, Lindsay Willmore, Andrew J. Ballard, Joshua Bambrick, Sebastian W. Bodenstein, David A. Evans, Chia-Chun Hung, Michael O’Neill, David Reiman, Kathryn Tunyasuvunakool, Zachary Wu, Akvile˙ Zemgulytˇ e, Eirini Arvan-˙ iti, Charles Beattie, Ottavia Bertolli, Alex Bridgland, Alexey Cherepanov, Miles Congreve, Alexander I. Cowen-Rivers, Andrew Cowie, Michael Figurnov, Fabian B. Fuchs, Hannah Gladman, Rishub Jain, Yousuf A. Khan, Caroline M. R. Low, Kuba Perlin, Anna Potapenko, Pascal Savy, Sukhdeep Singh, Adrian Stecula, Ashok Thillaisundaram, Catherine Tong, Sergei Yakneen, Ellen D. Zhong, Michal Zielinski, Augustin Zˇ´ıdek, Victor Bapst, Pushmeet Kohli, Max Jaderberg, Demis Hassabis, and John M. Jumper. Accurate structure prediction of biomolecular interactions with AlphaFold 3. Nature, pages 1–3, May 2024.

(4) Raktim Mitra, Jinsen Li, Jared M. Sagendorf, Yibei Jiang, Ari S. Cohen, Tsu-Pei Chiu, Cameron J. Glasscock, and Remo Rohs. Geometric deep learning of protein–DNA binding specificity. Nat Methods, 21(9):1674–1683, September 2024.

(5) Lin Yang, Yaron Orenstein, Arttu Jolma, Yimeng Yin, Jussi Taipale, Ron Shamir, and Remo Rohs. Transcription factor family-specific DNA shape readout revealed by quantitative specificity models. Mol Syst Biol, 13(2):910, February 2017.

(6) Takaya Saito and Marc Rehmsmeier. The Precision-Recall Plot Is More Informative than the ROC Plot When Evaluating Binary Classifiers on Imbalanced Datasets. PLoS ONE, 10(3):e0118432, March 2015.

(7) Mu Gao and Jeffrey Skolnick. DBD-Hunter: a knowledge-based method for the prediction of DNA-protein interactions. Nucleic Acids Res, 36(12):3978–3992, July 2008.

(8) H. Tomas Rube, Chaitanya Rastogi, Siqian Feng, Judith F. Kribelbauer, Allyson Li, Basheer Becerra, Lucas A. N. Melo, Bach Viet Do, Xiaoting Li, Hammaad H. Adam, Neel H. Shah, Richard S. Mann, and Harmen J. Bussemaker. Prediction of protein–ligand binding affinity from sequencing data with interpretable machine learning. Nat Biotechnol, 40(10):1520–1527, October 2022.

(9) Babak Alipanahi, Andrew Delong, Matthew T Weirauch, and Brendan J Frey. Predicting the sequence specificities of DNA- and RNA-binding proteins by deep learning. Nat Biotechnol, 33(8):831–838, August 2015.

(10) Joshua L. Wetzel, Kaiqian Zhang, and Mona Singh. Learning probabilistic proteinDNA recognition codes from DNA-binding specificities using structural mappings. Genome Res, 32(9):1776–1786, September 2022.

(11) Aziz Khan, Oriol Fornes, Arnaud Stigliani, Marius Gheorghe, Jaime A CastroMondragon, Robin van der Lee, Adrien Bessy, Jeanne Cheneby, Shubhada R Kulka-` rni, Ge Tan, Damir Baranasic, David J Arenillas, Albin Sandelin, Klaas Vandepoele, Boris Lenhard, Benoˆıt Ballester, Wyeth W Wasserman, Franc¸ois Parcy, and Anthony Mathelier. JASPAR 2018: update of the open-access database of transcription factor binding profiles and its web framework. Nucleic Acids Research, 46(D1):D260–D266, January 2018.

(12) Ivan V. Kulakovskiy, Ilya E. Vorontsov, Ivan S. Yevshin, Ruslan N. Sharipov, Alla D. Fedorova, Eugene I. Rumynskiy, Yulia A. Medvedeva, Arturo Magana-Mora, Vladimir B. Bajic, Dmitry A. Papatsenko, Fedor A. Kolpakov, and Vsevolod J. Makeev. HOCOMOCO: towards a complete collection of transcription factor binding models for human and mouse via large-scale ChIP-Seq analysis. Nucleic Acids Res, 46(D1):D252–D259, January 2018.

(13) Arttu Jolma, Jian Yan, Thomas Whitington, Jarkko Toivonen, Kazuhiro R. Nitta, Pasi Rastas, Ekaterina Morgunova, Martin Enge, Mikko Taipale, Gonghong Wei, Kimmo Palin, Juan M. Vaquerizas, Renaud Vincentelli, Nicholas M. Luscombe, Timothy R. Hughes, Patrick Lemaire, Esko Ukkonen, Teemu Kivioja, and Jussi Taipale. DNABinding Specificities of Human Transcription Factors. Cell, 152(1-2):327–339, January 2013.

(14) Maor Asif and Yaron Orenstein. DeepSELEX: inferring DNA-binding preferences from HT-SELEX data using multi-class CNNs. Bioinformatics, 36(Supplement 2):i634–i642, December 2020.

(15) Oriol Fornes, Alberto Meseguer, Joachim Aguirre-Plans, Patrick Gohl, Patricia M Bota, Ruben Molina-Fernandez, Jaume Bonet, Altair Chinchilla-Hernandez, Ferran´ Pegenaute, Oriol Gallego, Narcis Fernandez-Fuentes, and Baldo Oliva. Structurebased learning to predict and model protein–DNA interactions and transcriptionfactor co-operativity in cis -regulatory elements. NAR Genomics and Bioinformatics, 6(2):lqae068, April 2024.

(16) Sofia Aizenshtein-Gazit and Yaron Orenstein. DeepZF: improved DNA-binding prediction of C2H2-zinc-finger proteins by deep transfer learning. Bioinformatics, 38(Suppl 2):ii62–ii67, September 2022.

(17) Stephen K Burley, Charmi Bhikadiya, Chunxiao Bi, Sebastian Bittrich, Henry Chao, Li Chen, Paul A Craig, Gregg V Crichlow, Kenneth Dalenberg, Jose M Duarte, Shuchismita Dutta, Maryam Fayazi, Zukang Feng, Justin W Flatt, Sai Ganesan, Sutapa Ghosh, David S Goodsell, Rachel Kramer Green, Vladimir Guranovic, Jeremy Henry, Brian P Hudson, Igor Khokhriakov, Catherine L Lawson, Yuhe Liang, Robert Lowe, Ezra Peisach, Irina Persikova, Dennis W Piehl, Yana Rose, Andrej Sali, Joan Segura, Monica Sekharan, Chenghua Shao, Brinda Vallat, Maria Voigt, Ben Webb, John D Westbrook, Shamara Whetstone, Jasmine Y Young, Arthur Zalevsky, and Christine Zardecki. RCSB Protein Data Bank (RCSB.org): delivery of experimentally-determined PDB structures alongside one million computed structure models of proteins from artificial intelligence/machine learning. Nucleic Acids Research, 51(D1):D488–D508, November 2022.

(18) Raktim Mitra, Ari S. Cohen, Jared M. Sagendorf, Helen M. Berman, and Remo Rohs. DNAproDB: an updated database for the automated and interactive analysis of protein-DNA complexes. Nucleic Acids Res, 53(D1):D396–D402, January 2025.

(19) Natalia Petrenko, Yi Jin, Liguo Dong, Koon Ho Wong, and Kevin Struhl. Requirements for RNA polymerase II preinitiation complex formation in vivo. eLife, 8:e43654, January 2019.

(20) Rudolf Jaenisch and Adrian Bird. Epigenetic regulation of gene expression: how the genome integrates intrinsic and environmental signals. Nat Genet, 33(3):245–254, March 2003.

(21) Claire Marchal, Jiao Sima, and David M. Gilbert. Control of DNA replication timing in the 3D genome. Nat Rev Mol Cell Biol, 20(12):721–737, December 2019.

(22) Lucia A. Hindorff, Praveen Sethupathy, Heather A. Junkins, Erin M. Ramos, Jayashri P. Mehta, Francis S. Collins, and Teri A. Manolio. Potential etiologic and functional implications of genome-wide association loci for human diseases and traits. Proceedings of the National Academy of Sciences, 106(23):9362–9367, June 2009.

(23) Tuuli Lappalainen, Alexandra J Scott, Margot Brandt, and Ira M Hall. Genomic analysis in the age of human genome sequencing. Cell, 177(1):70–84, 2019.

(24) Sonali Mukherjee, Michael F. Berger, Ghil Jona, Xun S. Wang, Dale Muzzey, Michael Snyder, Richard A. Young, and Martha L. Bulyk. Rapid analysis of the DNA-binding specificities of transcription factors with DNA microarrays. Nat Genet, 36(12):1331– 1339, December 2004.

(25) Shaoxun Liu, Pilar Gomez-Alcala, Christ Leemans, William J. Glassford, Lucas A. N. Melo, Xiang-Jun Lu, Richard S. Mann, and Harmen J. Bussemaker. Predicting the DNA binding specificity of transcription factor mutants using family-level biophysically interpretable machine learning. bioRxiv, page 2024.01.24.577115, April 2025.

(26) Tsu-Pei Chiu, Satyanarayan Rao, and Remo Rohs. Physicochemical models of protein–DNA binding with standard and modified base pairs. Proc. Natl. Acad. Sci. U.S.A., 120(4):e2205796120, January 2023.

(27) Matthew T Weirauch, Atina Cote, Raquel Norel, Matti Annala, Yue Zhao, Todd R Riley, Julio Saez-Rodriguez, Thomas Cokelaer, Anastasia Vedenko, Shaheynoor Talukder, and others. Evaluation of methods for modeling transcription factor sequence specificity. Nature biotechnology, 31(2):126–134, 2013.

(28) Chaitanya Rastogi, H. Tomas Rube, Judith F. Kribelbauer, Justin Crocker, Ryan E. Loker, Gabriella D. Martini, Oleg Laptenko, William A. Freed-Pastor, Carol Prives, David L. Stern, Richard S. Mann, and Harmen J. Bussemaker. Accurate and sensitive quantification of protein-DNA binding affinity. Proc. Natl. Acad. Sci. U.S.A., 115(16), April 2018.

(29) Rahmatullah Roche, Bernard Moussad, Md Hossain Shuvo, Sumit Tarafder, and Debswapna Bhattacharya. EquiPNAS: improved protein–nucleic acid binding site prediction using protein-language-model-informed equivariant deep graph neural networks. Nucleic Acids Research, 52(5):e27–e27, March 2024.

(30) Yufan Liu and Boxue Tian. Protein–DNA binding sites prediction based on pretrained protein language model and contrastive learning. Briefings in Bioinformatics, 25(1):bbad488, November 2023.

(31) Binh P. Nguyen, Quang H. Nguyen, Giang-Nam Doan-Ngoc, Thanh-Hoang Nguyen-Vo, and Susanto Rahardja. iProDNA-CapsNet: identifying protein-DNA binding residues using capsule neural networks. BMC Bioinformatics, 20(S23):634, December 2019.

(32) Trevor Siggers and Raluca Gordan. Protein–DNA binding: complexities and multi-ˆ protein codes. Nucleic Acids Research, 42(4):2099–2111, February 2014.

(33) Johannes Soding, Andreas Biegert, and Andrei N. Lupas. The HHpred interactive¨ server for protein homology detection and structure prediction. Nucleic Acids Research, 33(suppl 2):W244–W248, July 2005.

(34) William Humphrey, Andrew Dalke, and Klaus Schulten. VMD – Visual Molecular Dynamics. Journal of Molecular Graphics, 14:33–38, 1996.

(35) Arttu Jolma, Teemu Kivioja, Jarkko Toivonen, Lu Cheng, Gonghong Wei, Martin Enge, Mikko Taipale, Juan M Vaquerizas, Jian Yan, Mikko J Sillanpa¨a, and others.¨ Multiplexed massively parallel SELEX for characterization of human transcription factor binding specificities. Genome research, 20(6):861–873, 2010.

(36) Nobuo Ogawa and Mark D Biggin. High-throughput SELEX determination of DNA sequences bound by transcription factors in vitro. Gene Regulatory Networks: Methods and Protocols, pages 51–63, 2012.

(37) Alina Isakova, Romain Groux, Michael Imbeault, Pernille Rainer, Daniel Alpern, Riccardo Dainese, Giovanna Ambrosini, Didier Trono, Philipp Bucher, and Bart Deplancke. SMiLE-seq identifies binding motifs of single and dimeric transcription factors. Nature methods, 14(3):316–322, 2017.

(38) Paul G. Giresi, Jonghwan Kim, Ryan M. McDaniell, Vishwanath R. Iyer, and Jason D. Lieb. FAIRE (Formaldehyde-Assisted Isolation of Regulatory Elements) isolates active regulatory elements from human chromatin. Genome Res., 17(6):877–885, January 2007.

(39) Peter J Park. ChIP–seq: advantages and challenges of a maturing technology. Nature reviews genetics, 10(10):669–680, 2009.

(40) Terrence S. Furey. ChIP–seq and beyond: new and improved methodologies to detect and characterize protein–DNA interactions. Nat Rev Genet, 13(12):840–852, December 2012.

(41) Anna Bartlett, Ronan C. O’Malley, Shao-shan Carol Huang, Mary Galli, Joseph R. Nery, Andrea Gallavotti, and Joseph R. Ecker. Mapping genome-wide transcriptionfactor binding sites using DAP-seq. Nat Protoc, 12(8):1659–1672, August 2017.

(42) Marcel Geertz, David Shore, and Sebastian J Maerkl. Massively parallel measurements of molecular interaction kinetics on a microfluidic platform. Proceedings of the National Academy of Sciences, 109(41):16540–16545, 2012.

(43) Gary D. Stormo and Yue Zhao. Determining the specificity of protein–DNA interactions. Nat Rev Genet, 11(11):751–760, November 2010.

(44) Xingcheng Lin, Rachel Leicher, Shixin Liu, and Bin Zhang. Cooperative DNA looping by PRC2 complexes. Nucleic Acids Research, 49(11):6238–6248, June 2021.

(45) P. L. Privalov, A. I. Dragan, and C. Crane-Robinson. Interpreting protein/DNA interactions: distinguishing specific from non-specific and electrostatic from nonelectrostatic components. Nucleic Acids Research, 39(7):2483–2491, April 2011.

(46) J D Bryngelson and P G Wolynes. Spin glasses and the statistical mechanics of protein folding. Proc. Natl. Acad. Sci. U.S.A., 84(21):7524–7528, November 1987.

(47) J. N. Onuchic, Z. Luthey-Schulten, and P. G. Wolynes. Theory of protein folding: the energy landscape perspective. Annu Rev Phys Chem, 48:545–600, 1997.

(48) N. P. Schafer, B. L. Kim, W. Zheng, and P. G. Wolynes. Learning To Fold Proteins Using Energy Landscape Theory. Isr J Chem, 54(8-9):1311–1337, August 2014.

(49) Wen-Ting Chu, Zhiqiang Yan, Xiakun Chu, Xiliang Zheng, Zuojia Liu, Li Xu, Kun Zhang, and Jin Wang. Physics of biomolecular recognition and conformational dynamics. Rep. Prog. Phys., 84(12):126601, December 2021.

(50) Sebastian J. Maerkl and Stephen R. Quake. A Systems Approach to Measuring the Binding Energy Landscapes of Transcription Factors. Science, 315(5809):233–237, January 2007.